# Mitochondrial protein import determines lifespan through metabolic reprogramming and de novo serine biosynthesis

Eirini Lionaki[1,6✉], Ilias Gkikas [1,2,6], Ioanna Daskalaki[1,2], Maria-Konstantina Ioannidi [3,4], Maria I. Klapa [3] & Nektarios Tavernarakis [1,5✉]

Sustained mitochondrial fitness relies on coordinated biogenesis and clearance. Both processes are regulated by constant targeting of proteins into the organelle. Thus, mitochondrial protein import sets the pace for mitochondrial abundance and function. However, our understanding of mitochondrial protein translocation as a regulator of longevity remains enigmatic. Here, we targeted the main protein import translocases and assessed their contribution to mitochondrial abundance and organismal physiology. We find that reduction in cellular mitochondrial load through mitochondrial protein import system suppression, referred to as MitoMISS, elicits a distinct longevity paradigm. We show that MitoMISS triggers the mitochondrial unfolded protein response, orchestrating an adaptive reprogramming of metabolism. Glycolysis and de novo serine biosynthesis are causatively linked to longevity, whilst mitochondrial chaperone induction is dispensable for lifespan extension. Our findings extent the pro-longevity role of UPR[mt] and provide insight, relevant to the metabolic alterations that promote or undermine survival and longevity.

[1] Institute of Molecular Biology and Biotechnology, Foundation for Research and Technology — Hellas, Heraklion GR-70013, Crete, Greece. [2] Department of Biology, School of Sciences and Engineering, University of Crete, Heraklion GR-70013, Crete, Greece. [3] Metabolic Engineering and Systems Biology Laboratory, Institute of Chemical Engineering Sciences, Foundation for Research and Technology-Hellas (FORTH/ICE-HT), Patras GR-26504, Greece. [4] Department of Biology, School of Natural Sciences, University of Patras, Patras GR-26500, Greece. [5] Division of Basic Sciences, School of Medicine, University of Crete, Heraklion GR-71003, Crete, Greece. [6] These authors contributed equally: Eirini Lionaki, Ilias Gkikas. ✉email: lionaki@imbb.forth.gr; tavernarakis@imbb.forth.gr

Longevity is intimately linked to mitochondrial function[1]. Inherently compromised mitochondria exist in patients of the so-called mitochondrial diseases, who carry mutations in nuclear or mitochondrial genes encoding mitochondrial proteins and RNA molecules. Dysfunctional mitochondria have been shown to accrue with age in several organisms and correlate with a plethora of late-onset life-threatening diseases, like neurodegenerative disorders, cardiovascular diseases, and cancer[2–5]. This relationship could be attributed to a higher pace of mutation accumulation on mitochondrial DNA (mtDNA), clonal expansion of mutated mtDNA, and age-dependent decline of organelle-specific quality control mechanisms[6–9]. Given the variety and severity of mitochondrial-associated diseases, preservation of mitochondrial function throughout aging has been a long-standing goal, with therapeutic efforts focusing mainly on the enhancement of mitochondrial biogenesis and quality control.

Mitochondrial biogenesis, function, and abundance entail a proper mitochondrial protein import system[10]. The mitochondrial protein import system mediates translocation of 99% of all mitochondrial proteins and is conserved from yeast to humans[11]. Compromised mitochondrial protein import efficiency leads to loss of cytosolic and mitochondrial homeostasis. To counteract the accumulation of unimported proteins, various cellular proteostatic responses are elicited both inside and outside mitochondria[12–18]. Interestingly, proteostatic signaling has been linked to longevity and stress adaptation possibly through the preservation of a functional proteome[19,20]. However, in the long run, chronic activation of such energy-demanding proteostatic pathways can backfire[21–24].

Mitochondrial protein import sets the pace for mitochondrial biogenesis across species[10]. Although, increased mitochondrial biogenesis ensures the building of a robust and healthy mitochondrial network, accumulation of dysfunctional organelles, which often accompanies aging and age-associated diseases, can be detrimental for organismal physiology. In this context attenuation of mitochondrial biogenesis could act protectively against the build-up of the dysfunction. Emerging evidence suggests that reduced mitochondrial abundance can be beneficial in old age and disease background. Specifically, reduction of mitochondrial load protects against cellular senescence, a process known to mediate age-related tissue deterioration[25,26]. Moreover, knock-out (KO) of mouse transcription factor A mitochondrial (TFAM), a transcription factor required for mitochondrial DNA expression and maintenance[27,28] protects mice from age and diet-induced obesity and insulin resistance[29,30]. Despite the evidence, reduction of mitochondrial load as a pro-longevity intervention has never been tested.

To directly assess whether the attenuation of mitochondrial network expansion could serve as an alternative mechanism for addressing age-associated decline we targeted the mitochondrial protein import system. We used the versatile nematode *C. elegans*, to assess the effects of reduced mitochondrial protein import on organismal physiology. We show that inhibition of the main routes for mitochondrial protein import, leading to mitochondrial mass reduction, activates the organellar proteostatic mechanism, mitochondrial unfolded protein response (UPR^mt), which in sequence, triggers a longevity paradigm that diverges from the established nutrient-sensing and mild mitochondrial dysfunction pathways. Interestingly, induction of UPR^mt associated chaperones critical for organellar proteostasis does not seem to be causatively linked to the observed phenotype. Combined metabolomic and genetic analysis provided significant leads on the mechanisms underlying this phenomenon, extending our understanding on the biological processes influencing aging and longevity.

## Results

**Mitochondrial protein import inhibition promotes longevity.** Import of the vast majority of mitochondrial proteins is mediated by four main translocase complexes, TOM, TIM23, TIM22, and SAM/TOB (Supplementary Fig. 1a). After passing through the translocase of the outer membrane (TOM), mitochondrial preproteins are delivered to different translocases depending on their final sub-mitochondrial destination. All matrix-targeted precursors and a large number of inner membrane-targeted preproteins are taken over by translocase of inner membrane 23 (TIM23). Finally, precursors of the outer membrane are inserted through the sorting and assembly machinery/translocase of the outer membrane for beta barrel proteins (SAM/TOB) complex, while targeting of polytopic inner membrane carrier preproteins is facilitated by the translocase of inner membrane 22 (TIM22)[10,11,31]. To specifically target mitochondrial mass, we genetically suppressed core components of the mitochondrial protein import machinery. Protein sequence similarity search identified the *C. elegans* homologs of the channel-forming subunits of the four main translocase complexes of the outer and inner mitochondrial membranes (Supplementary Fig. 1b). In this context, TOMM-40, TIMM-23, TIMM-22, and GOP-3, the *C. elegans* homologs of mammalian TOMM40, TIMM23, TIMM22 and SAMM50 respectively, were further examined. We genetically suppressed the aforementioned genes, by RNAi feeding of wild-type animals from hatching (Supplementary Fig. 1c). We found that genetic inhibition of *timm-23* and *tomm-40* caused phenotypic alterations typical for mitochondrial mutants, such as reduced body size (Supplementary Fig. 1d), reduced brood size (Supplementary Fig. 1e), and a slight developmental delay of 6–8 h. These phenotypic alterations were missing from the *timm-22* and *gop-3* deficient animals. Looking into the mitochondrial network of adult worms upon mitochondrial protein import impairment we noted significant modifications. Knocking down of all translocase subunits has a pronounced effect on mitochondrial morphology, as evidenced by the muscle- and intestine-specific mitochondrial reporters (Fig. 1a, b). To investigate the effects of each translocase on mitochondrial abundance and functionality we used a single copy translational reporter of cytochrome c oxidase subunit 4 (COX-4::GFP) as well as potential-dependent and-independent mitochondria-specific dyes (Fig. 1c–f)[32]. Finally, protein levels of ATP synthase subunit alpha (ATP-1) and mitochondrial to nuclear DNA ratio were examined (Fig. 1g and Supplementary Fig. 1g). As anticipated, knockdown of the main import translocases, *tomm-40* and *timm-23*, resulted in a prominent reduction of COX-4 and ATP-1 protein levels, mitochondrial membrane potential, ROS and ATP levels, arguing for a significant depletion of mitochondrial load. Conversely, *gop-3* and *timm-22* silencing cause a moderate reduction in COX-4 levels and membrane potential (Fig. 1c, e) and do not affect staining by the potential-independent dye Mitotracker Green or ATP-1 levels (Fig. 1d, g), while *timm-22* RNAi even leads to a slight increase in mitochondrial ROS production (Fig. 1f). Interestingly, the copy number of mitochondrial DNA is dramatically decreased upon inhibition of all mitochondrial protein import subunits (Supplementary Fig. 1g). These observations are justified considering that TOM40 is the general import pore for the organelle and TIM23 mediates translocation of approximately two-thirds of all mitochondrial proteins. Reduction in Mito-chondrial abundance driven by mitochondrial protein import system suppression, observed upon TOMM-40 or TIMM-23 depletion, will be referred to as MitoMISS.

The effects of MitoMISS on lifespan were investigated through survival analysis of otherwise wild type worms in which we downregulated the aforementioned mitochondrial protein

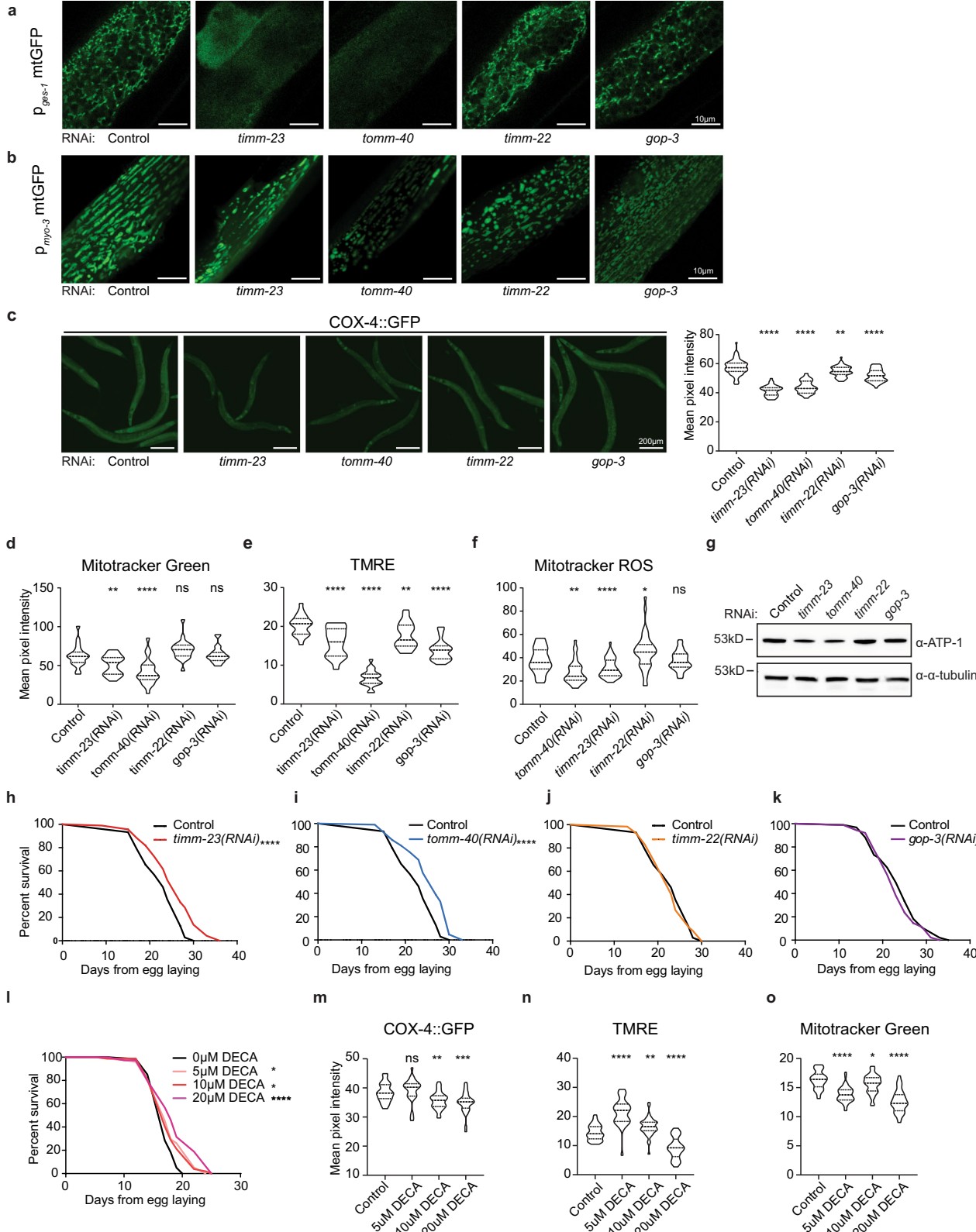

import-related genes. In agreement with our working hypothesis MitoMISS is beneficial for lifespan. Specifically, inhibition of *timm-23* or *tomm-40* increases lifespan, while suppression of *timm-22* or *gop-3*, that does not trigger MitoMISS, does not affect longevity (Fig. 1h–k). To assess the temporal requirements of MitoMISS longevity, we subjected wild type stage 4 larvae (L4) to RNAi against *timm-23* and *tomm-40* throughout adulthood. We

found that post-developmental inhibition of *tomm-40* and *timm-23*, moderately reduces mitochondrial abundance and function while it preserves part of its pro-longevity capacity (Supplementary Fig. 1h–k). Intriguingly, pharmacological inhibition of mitochondrial protein import system using dequalinium chloride (DECA) reproduces MitoMISS effects on mitochondrial load and function as well as longevity (Fig. 1i–o)[33]. Collectively, these

**Fig. 1 Mitochondrial protein import inhibition elicits morphological and bioenergetic adaptations of the mitochondrial network and regulates lifespan.**
**a**, **b** Transgenic animals expressing a mitochondrial-targeted GFP in the intestine (**a**) or body-wall muscles (**b**), were grown from hatching until day 1 of adulthood on the indicated RNAi expressing bacteria and then monitored by confocal microscopy. $n = 3$ biologically independent experiments with similar results. **c** Epifluorescence images depicting total COX-4 endogenous protein levels in animals grown from hatching until day 1 of adulthood on the indicated RNAi expressing bacteria (left panel) and COX-4::GFP fluorescence quantification in violin plot (right panel). $n = 3$ biologically independent experiments with at least 20 worms per condition. Exact sample size and $P$ values are included in Source Data file. **d**–**f** Day 1 adult animals were stained with Mito Tracker Green, which labels mitochondria independently of membrane potential (**d**), TMRE, which is a membrane potential-dependent mitochondrial dye (**e**), and Mito Tracker Red CM-H2Xros, which is a mitochondrial ROS specific dye (**f**). The violin plots represent the mean fluorescent intensity of at least 20 worms. All treatments were compared to control with two-tailed $t$-test. $n = 3$ biologically independent experiments with at least 20 worms. Exact sample size and $P$ values are included in the Source Data file. **g** Immunoblot analysis of total ATP-1 protein levels in animals grown from hatching until day 1 of adulthood on the indicated RNAi expressing bacteria. $n = 2$ biologically independent experiments. **h**–**k** Lifespan curves of control-treated and animals treated with *timm-23*(RNAi) (**h**), *tomm-40*(RNAi) (**i**), *timm-22*(RNAi) (**j**), and *gop-3*(RNAi) (**k**) from egg. $n = 3$ biologically independent experiments. **l** Lifespan curve of DECA-treated animals in various concentrations from the egg. $n = 2$ biologically independent experiments. Statistical analysis for lifespan curves was performed with the Log-rank (Mantel–Cox) test; detailed values are shown in Supplementary Table 2. **m** Quantification of COX-4::GFP fluorescence upon treatment with different DECA concentrations (depicted in violin plot). **n** Quantification of mitochondrial membrane Δψ (TMRE staining) upon treatment with different DECA concentrations. **o** Violin plot of quantified Mitotracker Green staining upon treatment with different DECA concentrations. **m**–**o** $n = 2$ biologically independent experiments. All treatments were compared to control with two-tailed $t$-test (**** denotes $P < 0.0001$, *** denotes $P < 0.001$, ** denotes $P < 0.01$ and * denotes $P < 0.05$) Exact sample size and $P$ values are included in Source Data file. a.u. arbitrary units.

findings underline that reduced mitochondrial load in otherwise wild type animals is sufficient to trigger longevity.

Diminished insulin/insulin-like growth factor 1 (IGF1) signaling and caloric restriction (CR) extend lifespan in multiple laboratory model animals like worms, yeast, flies, and mice, through modulation of the forkhead box transcription factors class O (FOXO)[34]. Therefore, *daf-2(e1370)* mutants (the *C. elegans* insulin/IGF1 receptor homolog), are exceptionally long-lived, while their longevity depends on the FOXO homolog, DAF-16. Interestingly, *timm-23* knockdown extends the lifespan of the long-lived *daf-2(e1370)* mutant, the short-lived *daf-16(mu86)* mutant, and the feeding-defective *eat-2(ad465)* mutant (Fig. 2a–c). Therefore, MitoMISS triggers an independent longevity pathway that acts in parallel to established nutrient availability-related pathways, like reduced insulin signaling and CR.

Low-insulin signaling has been proposed to promote longevity through a transient mitochondrial ROS signal that activates a mitohormetic response governed by SKiNhead-1 (SKN-1), the *C. elegans* homolog of nuclear factor-erythroid 2-related factor 2 (NRF2), and the nutrient-sensing AMP-activated kinase homolog (AAK-2)[35]. Interestingly, SKN-1 transcriptional activity is not induced under conditions that reduce mitochondrial mass and ROS levels, as shown by expression levels of its target gene glutathione S transferase (*gst-4*) (Fig. 2d). Moreover, both SKN-1 and AAK-2 are dispensable for lifespan extension upon *timm-23* suppression (Fig. 2e, f and Supplementary Fig. 1a, b). Interestingly, AAK-2 limits the pro-longevity effect of reduced mitochondrial mass, since *timm-23* knockdown in the *aak-2* mutant background further extended lifespan compared to their control counterparts. Conclusively, activation of a ROS-mediated oxidative stress or a mitohormetic response could not justify the beneficial effects of MitoMISS on lifespan.

Mild inhibition of mitochondrial respiration is known to extend lifespan in worms, flies, and mammals[36–39]. To assess whether the lifespan promoting effects of MitoMISS are due to respiration perturbations, we simultaneously knocked down *timm-23* with either *cco-1*, *atp-3*, or *ucr-1*. In all cases, MitoMISS further extends the lifespan of animals that experience mild mitochondrial dysfunction (Fig. 2g, h and Supplementary Fig. 1c). Moreover, homeobox transcription factor (CEH-23) and p53-like transcription factor (CEP-1) known to mediate the longevity effects of mitochondrial mutants and hypoxia-inducible factor (HIF-1) are dispensable for MitoMISS-linked longevity (Supplementary Fig. 2d–f)[40–43]. The latter suggests that MitoMISS acts in

parallel to mild mitochondrial dysfunction to promote longevity. Taken together, our findings show that MitoMISS mediates longevity synergistically with the traditional pathways of caloric restriction, low-insulin signaling, and mild mitochondrial stress.

**Components of UPR^mt are activated and required for MitoMISS-associated longevity.** Several studies have reported that acute mitochondrial protein import perturbations lead to increased proteotoxic burden in the cytoplasm, and thus trigger a retrograde signaling response which entails increased cytoplasmic chaperones and proteasome activity among others, safeguarding the proteome outside mitochondria[44,45]. To assess whether inhibition of protein import induces proteotoxic stress in the ER or the cytosol, transcriptional expression of proteasomal, UPR^ER, and cytosolic chaperones, as well as protein levels of the heat shock factor 1 (HSF-1), were examined. Interestingly, loss of mitochondrial protein import efficiency failed to induce the expression of the aforementioned proteostasis markers (Supplementary Fig. 3a–d). Phosphorylation of the eukaryotic translation initiation factor 2 alpha (eIF2α) leads to systemic protein translation inhibition which supports recovery from proteotoxic stress[46]. We find that although all protein import perturbations increase the phosphorylated form of eIF2α (they also increase its total form, therefore the ratio phospho/total levels of eIF2α does not increase), (Supplementary Fig. 3e). The latter corroborates the lack of a robust proteostatic stress response upon mitochondrial protein import deficiency. Our findings are in agreement with previous studies in *C. elegans*, which have shown that mitochondrial perturbations that affect the protein import machinery do not induce robust proteotoxic effects in the cytoplasm[19,47].

Mitochondrial proteotoxic stress can be partly alleviated by induction of the mitochondrial unfolded protein response pathway (UPR^mt). UPR^mt encompasses the transcriptional activation of an arsenal of proteases and chaperones that target mitochondria to relieve the proteotoxic burden. UPR^mt induction and the transcription factors driving it, DVE-1 and ATFS-1, have been proposed to mediate the beneficial effects of mild mitochondrial dysfunction on lifespan[48,49]. However, this notion has recently been challenged, since the induction of UPR^mt does not consistently correlate with increased lifespan while constitutive UPR^mt activation is not sufficient to trigger lifespan extension and can even be toxic[24,50]. To directly assess the involvement of UPR^mt in the longevity pathway triggered upon MitoMISS, we measured the transcriptional activation of two mitochondrial chaperones, *hsp-60* and *hsp-6*, that are known to be specifically

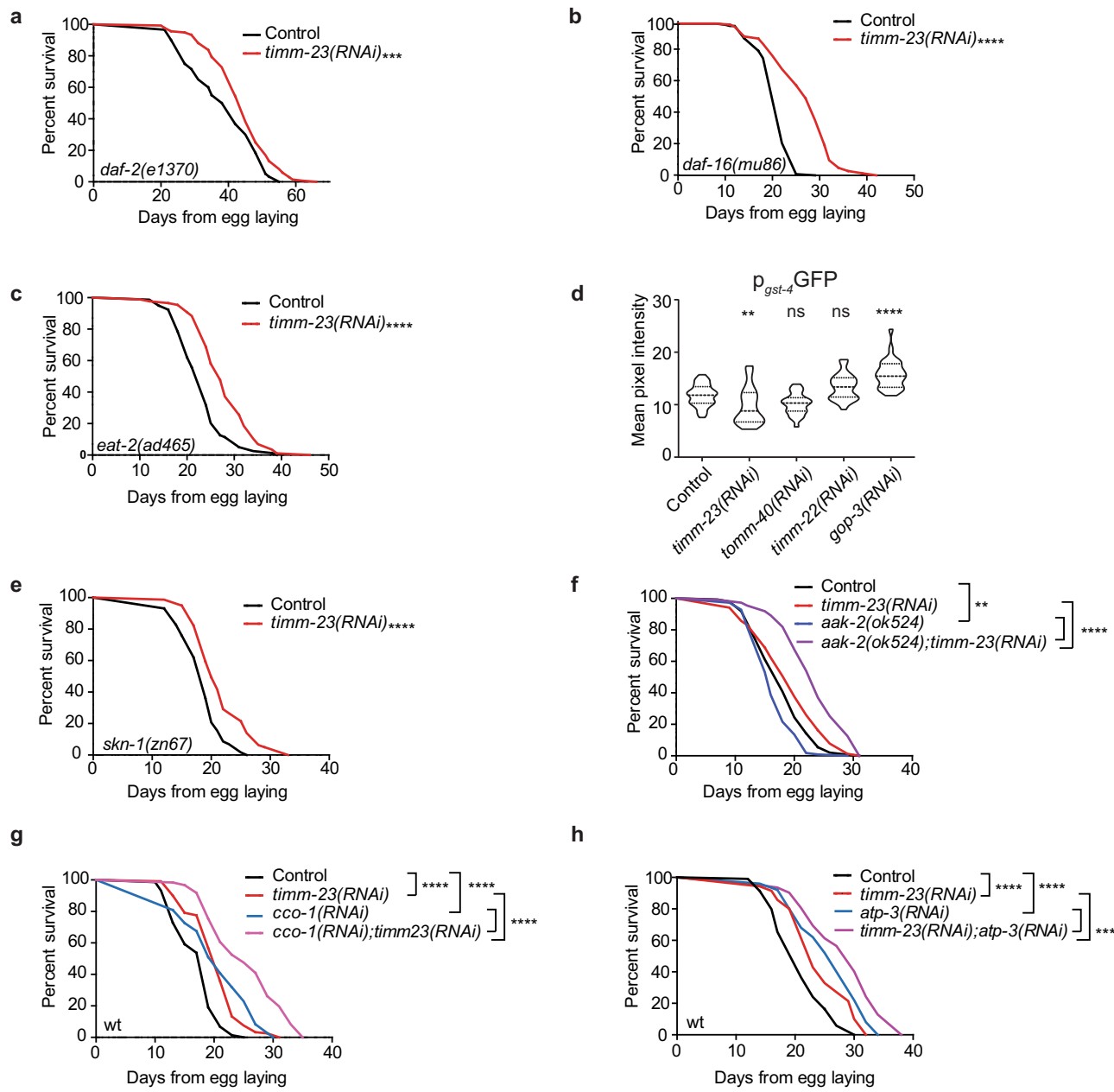

**Fig. 2 MitoMISS represents a longevity paradigm discrete from low-insulin signaling, caloric restriction, and mild mitochondrial stress. a–c** Lifespan curves of low-insulin and caloric restriction mutants, *daf-16, daf-2,* and *eat-2,* respectively, upon MitoMISS, **d** Expression levels of *gst-4* upon inhibition of mitochondrial protein import components, *timm-23, tomm-40, timm-22* and *gop-3* (Fluorescence intensities are depicted in violin plot). One-way ANOVA, with Dunnett's multiple comparisons test. **e, f** Lifespan curves of *skn-1* and *aak-2* mutants involved in mitohormesis, detoxification, and nutrient-sensing, respectively, upon MitoMISS. **g, h** Lifespan curves of *cco-1* and *atp-3* respiration-deficient animals upon MitoMISS. Statistical analysis for lifespan curves was performed with the Log-rank (Mantel–Cox) test; detailed values are shown in Supplementary Table 2 (**** denotes $P < 0.0001$, *** denotes $P < 0.001$, ** denotes $P < 0.01$ and * denotes $P < 0.05$).

induced by UPR$^{mt}$[51]. We assessed the expression levels of *hsp-60* and *hsp-6* upon both the lifespan-extending (*timm-23* and *tomm-40* knockdown) and -non extending (*timm-22* and *gop-3* knockdown) conditions. We find that all conditions activate UPR$^{mt}$, although MitoMISS triggers a more pronounced effect (Fig. 3a, b). Interestingly, post-developmental inhibition of *tomm-40* and *timm-23* fails to activate UPR$^{mt}$, in contrast to other known UPR$^{mt}$ triggering interventions (Supplementary Fig. 4a). To specifically test whether the additive longevity phenotype of *cco-1* and *atp-3* with *timm-23* inhibition is due to an additive UPR$^{mt}$ induction, we analyzed *hsp-6* expression in the conditions used in the lifespan experiment (Fig. 2g, h). Interestingly, we found that

*hsp-6* expression upon MitoMISS is not further induced by *cco-1* and *atp-3* suppression (Fig. 2g, h and Supplementary Fig. 4b). The aforementioned findings infer that MitoMISS-associated longevity may not be causatively linked to mitochondrial chaperone expression.

Consistent with literature evidence, both ATFS-1 and DVE-1 are required for UPR$^{mt}$ induction upon MitoMISS, as *hsp-6* cannot be induced when *atfs-1* or *dve-1* are inhibited (Fig. 3c, d and Supplementary Fig. 4c, d), or in an *atfs-1(tm4525)* or *dve-1(tm4803)* mutant background (Supplementary Fig. 4f, h). Subsequently, we pursued the involvement of ATFS-1 and DVE-1 in the observed longevity phenotype. Notably, knockdown

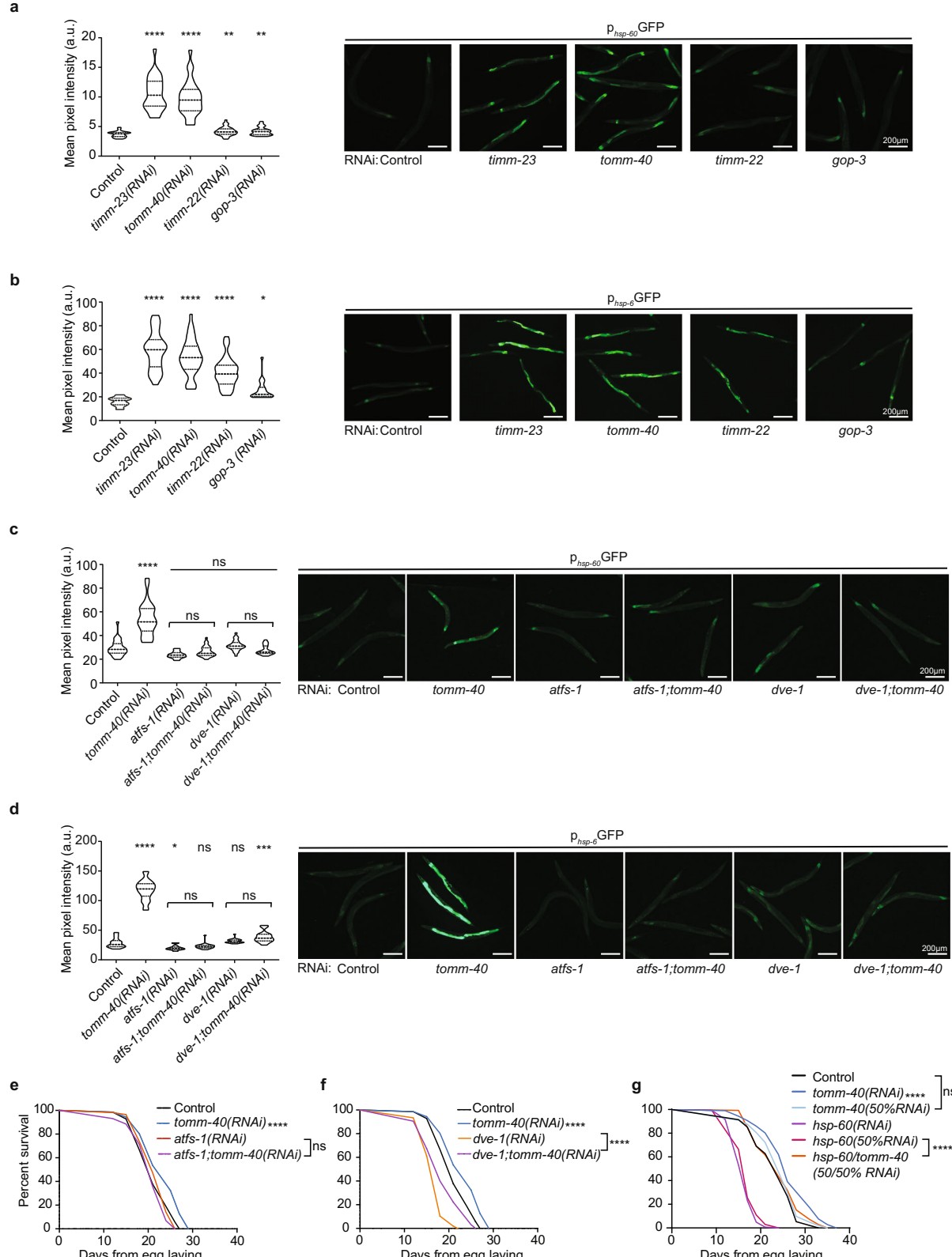

of either *timm-23* or *tomm-40* in the *atfs-1* null mutant background failed to phenocopy the longevity effect of MitoMISS (Supplementary Fig. 4e). Moreover, simultaneous inhibition of *tomm-40* and *atfs-1* abolishes MitoMISS-associated longevity in otherwise wild type animals (Fig. 3e). Interestingly, DVE-1 is neither induced upon MitoMISS nor required for longevity (Supplementary Fig. 4f, g, k and Fig. 3f). Finally, the peptide transporter HAF-1 is dispensable both for the chaperone induction and lifespan extension upon MitoMISS (Supplementary Fig. 4i, j)[52].

To further assess the necessity of UPR^mt components for lifespan extension we tested whether HSP-60 chaperonin is required for MitoMISS longevity. HSP-60 is implicated in the folding and stability of matrix-targeted mitochondrial proteins. In

**Fig. 3 MitoMISS longevity depends on ATFS-1 and is independent of UPR^mt-induced chaperones. a**, **b** Expression levels of canonical UPR^mt specific-transcriptional reporters *hsp-60* (**a**) and *hsp-6* (**b**) were monitored upon inhibition of mitochondrial protein import components, *timm-23*, *tomm-40*, *timm-22* and *gop-3*. $n = 3$ biologically independent experiments with at least 20 worms per condition. One-way ANOVA with Dunnett's multiple comparison test. Exact sample size and P values are included in the Source Data file. **c**, **d** Expression levels of *hsp-60* (**c**) and *hsp-6* (**d**) upon MitoMISS and in the absence of ATFS-1 or DVE-1. $n = 3$ biologically independent experiments with at least 20 worms per condition. One-way ANOVA with Tukey's multiple comparison test. Exact sample size and P values are included in the Source Data file. a.u. arbitrary units. **e**, **f** Lifespan curves upon MitoMISS in the absence and presence of ATFS-1 (**e**) and DVE-1 (**f**). **g** Lifespan curves upon MitoMISS in the absence and presence of HSP-60. $n = 2$ biologically independent experiments. Lifespan curves were statistically analyzed with the Log-rank (Mantel–Cox) test; detailed values are shown in Supplementary Table 2. (**** denotes $P < 0.0001$, *** denotes $P < 0.001$,** denotes $P < 0.01$ and * denotes $P < 0.05$).

nematodes, it is required for embryonic and larval development therefore mutants are lethal or sterile. RNAi suppression of *hsp-60* is known to reduce the lifespan of nematodes[50]. Consistently, we found that *hsp-60* inhibition from hatching significantly reduced the lifespan of worms (Fig. 3g). Surprisingly, MitoMISS completely rescues the phenotype of *hsp-60* deficient animals (Fig. 3g). These findings indicate that MitoMISS-associated longevity may not require the conventional players of UPR^mt signaling. More importantly, these data imply that mitochondrial proteostasis (*hsp-60* induction) is not the critical factor mediating MitoMISS longevity. On the contrary, MitoMISS can rescue animals from severe mitochondrial stress arising from their depletion.

Given that MitoMISS longevity is dependent on ATFS-1 but not on other UPR^mt signaling components, we sought to identify whether MitoMISS elicits a similar transcriptional response with conventional UPR^mt stressors. Among them, inhibition of *spg-7* represents a well-documented example of UPR^mt induction[18,53]. Therefore, expression levels of ATFS-1-dependent genes involved in mitochondrial dynamics, TCA, ETC, detoxification and glycolysis were examined upon *spg-7* knockdown. We analyzed different transcriptional and/or translational reporters of ATFS-1 target genes upon *spg-7* and *tomm-40* RNAi (Supplementary Fig. 5a–i)[53]. Relative expression of additional ATFS-1 targets was monitored upon the same conditions (Supplementary Fig. 5j). Notably, silencing of *tomm-40* and *spg-7* engenders a different transcriptional outcome. Overall, TCA genes (*cts-1*, *mdh-2*, *aco-1*, *icl-1*) emerged as differentially regulated upon *spg-7* and *tomm-40* RNAi, as *spg-7* suppression leads to activation of TCA genes whilst, MitoMISS leaves TCA genes unaffected or with a tendency towards reduction. Genes involved in glycolysis (*gpi-1, gpd-2, gpd-3, enol-1, aldo-1*) are induced in both conditions, with MitoMISS activating a more robust response than *spg-7* RNAi. Conclusively, the aforementioned findings suggest that the metabolic shift initiated by MitoMISS-related UPR^mt is the critical factor for MitoMISS-associated longevity.

**MitoMISS longevity depends on enhanced glucose uptake and glycolysis.** Consistently with the literature, we previously showed (Supplementary Fig. 5), that glycolytic genes are induced upon both *spg-7* inhibition and MitoMISS. Glycolysis induction is known to be governed by UPR^mt. As all mitochondrial protein import perturbations induce UPR^mt, in terms of mitochondrial chaperone induction, we sought to identify whether they all lead to induction of glycolytic gene expression. Therefore, we analyzed the transcriptional reporters for the putative glucose-6-phosphate isomerase, *gpi-1*, and the glyceraldehyde-3-phosphate dehydrogenase, *gpd-2*, upon *timm-23*, *tomm-40*, *timm-22*, and *gop-3* inhibition (Fig. 4a and Supplementary Fig. 6c). Notably, neither *timm-22* nor *gop-3* knockdown induced *gpi-1* or *gpd-2* expression despite an efficient *hsp-6* and *hsp-60* induction (Fig. 3a, b). Consistently, induction of *gpi-1* and *gpd-2* upon MitoMISS is ATFS-1-dependent (Supplementary Figs. 5a and 6b). Interestingly, the expression of other genes involved in metabolic pathways parallel to glycolysis such as glycogen metabolism and the

pentose phosphate pathway are also increased upon MitoMISS (Supplementary Fig. 6b).

To assess whether this induction can be conducive to longevity we simultaneously inhibited *gpi-1* and *tomm-40* and tested the longevity of this population compared to single *gpi-1*- and *tomm-40*-deficient worms. Interestingly, although *gpi-1* RNAi promotes lifespan, when combined with MitoMISS, the associated longevity of both interventions is lost and the lifespan of double RNAi-treated animals returns to wild type levels (Fig. 4b and Supplementary Fig. 6a). Collectively, our data couple the pro-longevity function of ATFS-1 to the concurrent metabolic shift, rather than the regulation of the mitochondrial chaperones.

We next reasoned that the observed increased glycolysis upon MitoMISS could be triggered by enhanced glucose uptake. To this end, we subjected wild type animals and *atfs-1* mutants, treated with control or *tomm-40* RNAi, to 2-NBDG, a fluorescent glucose analog. We found that reduction of mitochondrial mass triggers an enhancement of glucose uptake by wild type worms but not by *atfs-1* mutants (Fig. 4c). To ensure that the fluorescent signal we monitored derives from incorporated 2-NBDG and not from extracellular fluorescent material in the intestinal lumen, we performed the same analysis in wild type animals grown on a medium supplemented with D-glucose. D-glucose antagonizes with 2-NBDG for intracellular incorporation. As evidenced, worms grown on D-glucose have reduced 2-NBDG signal. To further verify the effect of MitoMISS on glucose levels, we generated transgenic animals expressing Glifon4000, a fluorescent glucose indicator, in pharyngeal muscles[54]. As expected, MitoMISS-treated animals exhibited high levels of glucose (Fig. 4d). Facilitated Glucose Transporter-isoform 1 (FGT-1) is the sole GLUT2-like protein out of eight predicted GLUT-like proteins reported to have glucose transporter activity in *C. elegans*[55]. To test whether FGT-1 mediates glucose incorporation upon MitoMiss, we treated *fgt-1(tm3165)* mutants with control and *tomm-40* RNAi bacteria and then tested for 2-NBDG incorporation. We noted that 2-NBDG incorporation is significantly reduced in the *fgt-1* mutant compared to control (Fig. 4c). Nevertheless, TOMM-40 depletion can still induce glucose uptake in the absence of FGT-1. The latter suggests that FGT-1 partially mediates the enhanced glucose uptake observed upon MitoMISS. Interestingly, *fgt-1* is transcriptionally upregulated in *tomm-40*-deficient worms, in an ATFS-1-dependent manner (Fig. 4e and Supplementary Fig. 6d). It was shown that *fgt-1* knockdown in wild type worms reduces glucose incorporation and metabolism and increases lifespan due to low insulin signaling[55]. Consistently, we noticed that *fgt-1* mutants are long-lived compared to wild type animals. However, genetic inhibition of *tomm-40* could not significantly extend the lifespan of *fgt-1* mutants, implying that glucose incorporation is required for the longevity phenotype induced by MitoMISS (Supplementary Fig. 6e).

**De novo serine biosynthesis mediates MitoMISS-associated longevity.** To further characterize the metabolic adaptations triggered by MitoMISS, we performed metabolic profiling of wild

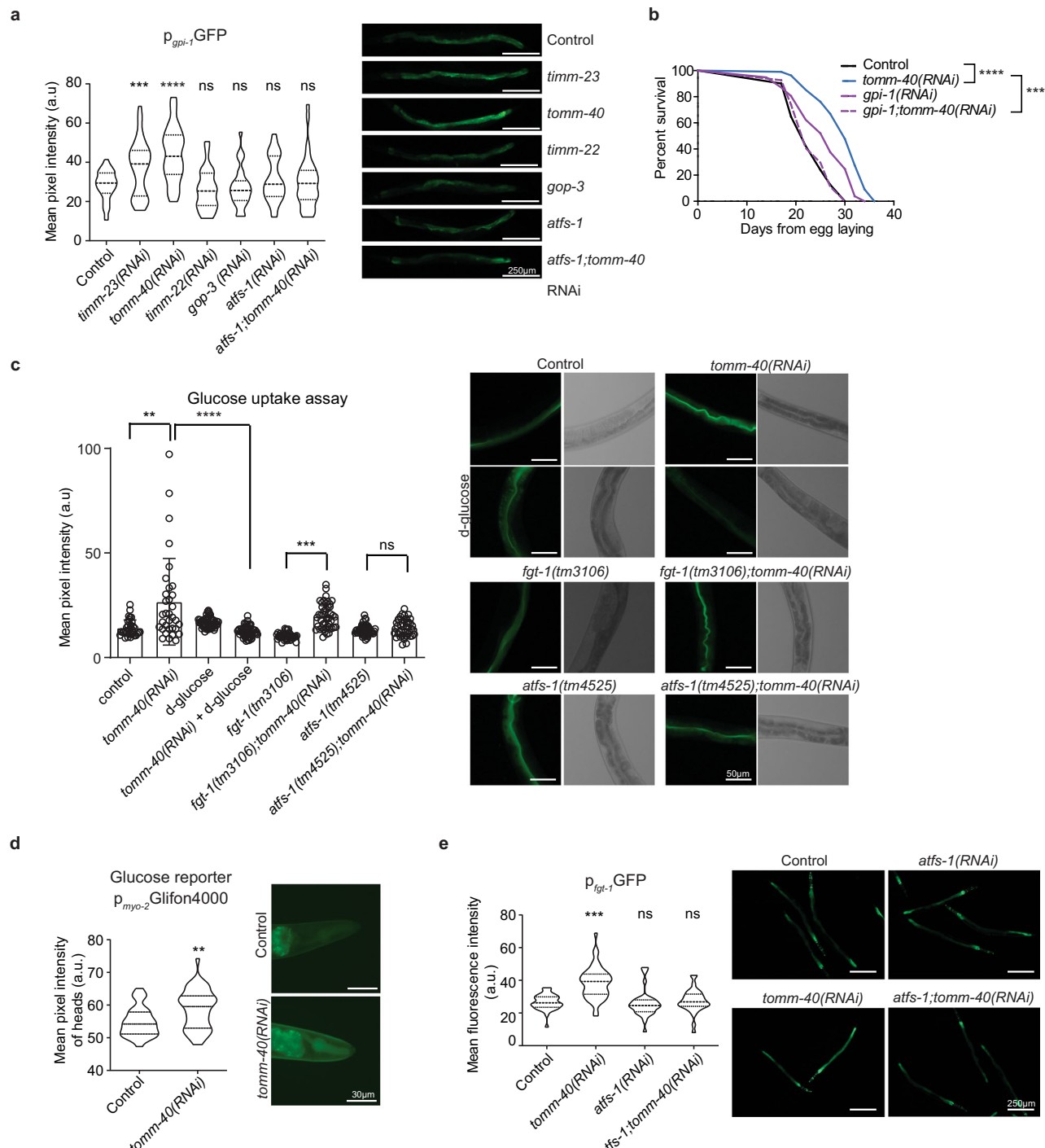

**Fig. 4 MitoMISS-induced longevity requires both glucose uptake and glycolysis in an ATFS-1-dependent manner. a** Expression levels of the glycolysis transcriptional reporter, *gpi-1*, upon inhibition of the mitochondrial protein import components, *timm-23*, *tomm-40*, *timm-22* and *gop-3*. *n* = 3 biologically independent experiments with at least 20 worms per condition. All conditions were compared to control with two-tailed *t*-test. **b** Lifespan curves of wild type animals in *gpi-1* deficient animals and upon MitoMISS. *n* = 3 biologically independent experiments. Lifespan curves were statistically analyzed with the Log-rank (Mantel–Cox) test. **c** Glucose uptake assay, using the fluorescent glucose analog 2-NBDG, upon MitoMISS. *n* = 3 biologically independent experiments with at least 20 worms per condition. Two-tailed *t*-test was used for comparisons. Data presented as mean ± SEM. **d** In vivo imaging of glucose levels of animals expressing Glifon4000 under the *myo-2* promoter upon MitoMISS. Fluorescence data depicted in violin plot. *n* = 2 biologically independent experiment. Comparison with two-tailed *t*-test. **e** Expression levels of the transcriptional reporter of glucose transporter *fgt-1* upon MitoMISS. Data depicted in violin plot. Fluorescent intensity of each sample was compared to the control with a two-tailed *t*-test. (**** denotes *P* < 0.0001, *** denotes *P* < 0.001,** denotes *P* < 0.01 and * denotes *P* < 0.05). Exact sample size and *P* values are included in the Source Data file. a.u. arbitrary units.

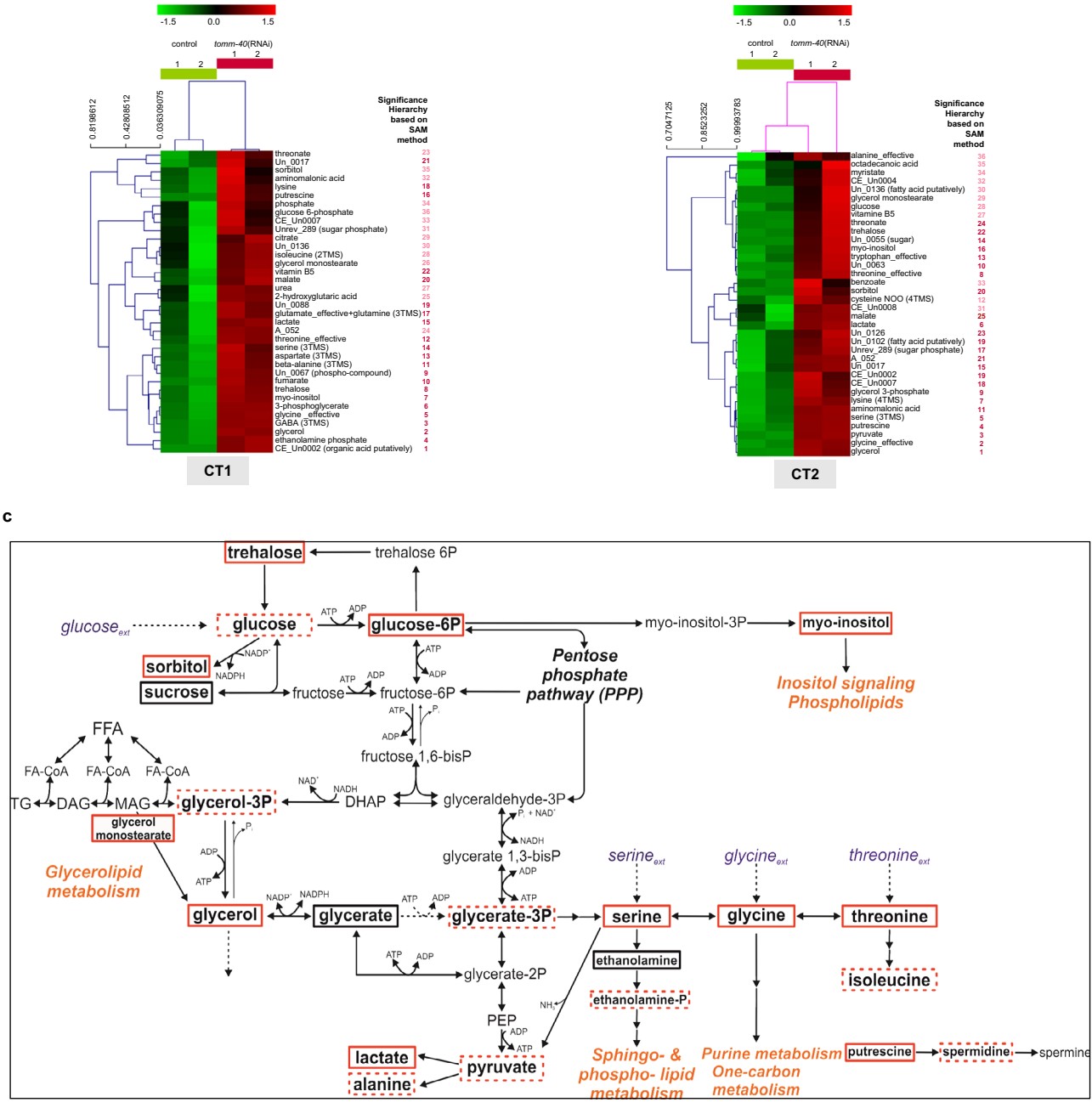

**Fig. 5 Metabolic profiling of control and MitoMISS animals. a**, **b** The hierarchical tree of the standardized abundance profiles of the positively significant metabolites in *tomm-40* RNAi worms compared to the controls based on Euclidean distance in CT1 (**a**) and CT2 (**b**). The color-code shows in green and red color, respectively, the abundances of a metabolite, if below or above its mean abundance in all profiles of the particular collection time data set (Supplementary Data File 1). The numbers next to each metabolite name indicate the order of significance of the metabolites based on the SAM method (see SAM curves in Supplementary Figs. 7c and 8c and full significant metabolite list in Supplementary Table 3). The red-colored numbers depict the significant metabolites identified at a significance threshold corresponding to FDR-median equal to zero and the pink-colored numbers depict the additional metabolites identified at the most lenient significance threshold possible for this data set, still corresponding at an FDR-median smaller than 1.5%. **c** The differences in the worm metabolic network from glucose to lactate in the presence and absence of *tomm-40*. All positively significant metabolites in both CT1 and CT2 as shown in Fig. 5a, b and Supplementary Table 3 are shown in red boxes. Profiled metabolites with no significant change in their abundance are shown in black boxes; the metabolites identified as positively significant in one of the two independent experiments, are shown in dashed red boxes.

type and *tomm-40* RNAi-treated animals in two independent experiments and collection times (Fig. 5a–c and Supplementary Data File 1, Supplementary Table 3, Supplementary Figs. 7, 8). The heatmaps of the metabolomic datasets support the distinct metabolic profile in the presence or absence of *tomm-40* (Fig. 5a, b and Supplementary Data File 1, Supplementary Figs. 7, 8). The significant differences (according to the SAM method) in the primary metabolism up to lactate in control and *tomm-40* RNAi-treated worms, are shown based on the differential metabolites visualized in the context of the relevant metabolic

network (Supplementary Table 3). Metabolites with significantly higher concentration upon *tomm-40* inhibition are shown in red boxes (dashed red boxes indicate increased abundance in these conditions that cannot be considered statistically significant due to biological variation between replicate samples) (Fig. 5c). Combining the differential metabolite information, we monitor an increased concentration of glycerate-3-phosphate (glycerate-3P) and other downstream metabolites. In this context, the metabolic profile agrees with the results from the *gpi-1* inhibition discussed in the previous section. Metabolic profiling supports the substantially higher glucose concentration upon *tomm-40* inhibition, which is accompanied by a higher abundance of trehalose, glucose-6-phosphate (glucose-6P), and sorbitol (Fig. 5c). These findings elucidate that the glucose-trehalose cycle is significantly enhanced upon MitoMISS, in agreement with a recent study that correlated this cycle with longevity[56]. These results also imply that part of the observed higher glucose uptake upon MitoMISS could be routed towards this cycle. Increased glycolysis, as supported by the increased concentration of glycerate-3P, appears responsible for the increased abundance of glycerol-3-phosphate (glycerol-3P), glycerol, and glycerol-monostearate. Through that pool, increased glycolysis is associated with another MitoMISS-related metabolic adaptation, the enhancement of the glycerolipid/free fatty acid (GL/FFA) cycle. Glycerol actually exhibits one of the two highest increases in abundance in the absence of *tomm40* at both collection times. The increased glycerol-3P observed under MitoMISS is expected to be synthesized from glycolysis and not from the phosphoenolpyruvate carboxykinase (PEPCK) transforming oxaloacetate to PEP under mainly gluconeogenic conditions[57]. Interestingly, this is further suggested by decreased mRNA and protein levels of PCK-2 (encoding *C. elegans* PEPCK) upon MitoMISS and independently of ATFS-1 (Supplementary Fig. 9a, b). In addition, metabolic routes out of the tricarboxylic acid cycle intermediates are expected with reduced flux under MitoMiss (Supplementary Fig. 5f, g). The GL/FFA cycle is considered "futile", because of the perpetuous synthesis and lysis of glycerolipids accompanied by heat release, with substantial ATP consumption. However, it has been associated with crucial intra- and inter-tissue signaling processes, including insulin sensitivity and vital lipid detoxification mechanisms, linked to longevity[58]. In accordance with the speculated increase in the GL/FFA cycle, we observed that the total levels of ATGL-1 lipase increase upon MitoMISS, although this increase does not depend on ATFS-1 (Supplementary Fig. 9c, d). On the other hand, we found that the total stored fat is increased upon MitoMISS, an effect that is reversed in *atfs-1* mutant animals (Supplementary Fig. 9e). This increase agrees with the increased glycero-phospholipid pools that are also revealed by the metabolic profiling. To differentiate between the various classes of lipids and/or tissues, further analysis is required. Overall, these results suggest a mobilization of lipid stores that coincides with increased lifespan. Moreover, the observed increased abundance of lactate upon MitoMISS, which may be due to increased replenishment from glycerol through the final glycolytic steps and/or the increased concentration of serine that can be directly transformed to pyruvate, further support the decreased mitochondrial abundance and thus activity upon MitoMISS.

A major observation acquired from metabolic profiling analysis is the association of the increased glycolysis rate upon MitoMISS with higher activity of de novo serine biosynthesis, through the increased abundance of glycerate-3P and serine. The high activity of de novo serine biosynthesis from glycolysis upon MitoMISS is further supported by the significantly higher abundance of glycine and threonine metabolite pools in the context of the rest of the changes detected. Reduction of the mitochondrial load and

activity would redirect metabolism towards alternative metabolic routes in the cytosol, especially those with a mitochondrial supplement, to assist in metabolic homeostasis and redox balance, compensating for the decrease in the net mitochondrial activity. This metabolic shift is supported by our measurements.

Serine can be incorporated from food or synthesized de novo from the glycolytic intermediate, glycerate-3P. The first and rate-limiting step in this biosynthetic reaction is catalyzed by phosphoglycerate dehydrogenase (PHGDH) in humans. The closest ortholog of this gene in *C. elegans* is *C31C9.2* which from now on will be referred to as *phgdh-1*. To assess the effects of MitoMISS on PHGDH protein levels we constructed a translational reporter of PHGDH-1 fused with GFP under its endogenous promoter. In agreement with the metabolomic data analysis, we found that MitoMISS increases PHGDH-1 protein levels (Fig. 6a). To test whether shuttling of carbon sources through glycolysis to de novo serine biosynthesis is required for lifespan extension; genetic inhibition of *phgdh-1* was performed in the presence or absence of TOMM-40 (Fig. 6b). Interestingly, we found that simultaneous silencing of *phgdh-1* and *tomm-40* completely abolished the longevity phenotype of the single RNAi treatments (Fig. 6b). Conversely, *phgdh-1* suppression had minimal effects on mitochondria, as evidenced by COX-4::GFP abundance and TMRE staining (Supplementary Fig. 10a–c). Moreover, TOMM-40 depletion recapitulates the MitoMISS phenotype with respect to mitochondrial abundance and function, even upon *phgdh-1* RNAi, despite the fact that it loses its beneficial effects on lifespan (Supplementary Fig. 10a–c). These data are in agreement with the metabolic profiling results and confirm that de novo serine biosynthesis is required for the MitoMISS longevity.

Since serine supplementation is known to increase lifespan in wild type nematodes[59], we sought to determine whether MitoMISS and exogenous serine trigger longevity through the same genetic pathway. In this context, we performed lifespan analysis of control and *tomm-40* RNAi-treated animals in the presence or absence of 5 mM serine. We noted that serine can increase the lifespan of wild type but not of *tomm-40* RNAi-treated worms (Fig. 6c). Next, we examined whether serine supplementation could substitute for de novo serine synthesis, following the lifespan of *phgdh-1* and *phgdh-1;tomm-40* RNAi-treated worms upon 5 mM serine supplementation. Interestingly, we found that exogenous serine could only partly rescue the longevity of the double *phgdh-1;tomm-40* RNAi-treated animals, suggesting that its mainly de novo serine synthesis rather than serine uptake that mediates MitoMISS longevity (Fig. 6d).

To further investigate the role of the exogenous serine supplementation as opposed to de novo serine biosynthesis on longevity, we exposed wild type as well as *tomm-40* and *phgdh-1* silenced animals to increasing concentrations of exogenous serine (5, 25, and 50 mM) from hatching throughout adulthood. Interestingly, we noted that in all genetic backgrounds increased concentrations of serine beyond 5 mM are detrimental for lifespan (Fig. 6e–g). TOMM-40 depleted animals are slightly more sensitive to increased concentrations of serine, likely due to the fact that their endogenous serine levels are already increased. Interestingly, *phgdh-1* deficient animals are not protected against high concentrations of exogenous serine. The latter suggests that, upon inhibition of de novo serine biosynthesis, endogenous serine levels remain stable, likely due to adaptation of dietary serine uptake.

The lifespan extension observed upon glycine and serine supplementation is dependent on the methionine cycle and S-adenosylmethionine (SAM) synthetase *sams-1*[60]. To test whether *sams-1* is required for MitoMiss-induced longevity we assessed the lifespan of *sams-1(ok2946)* mutant animals upon

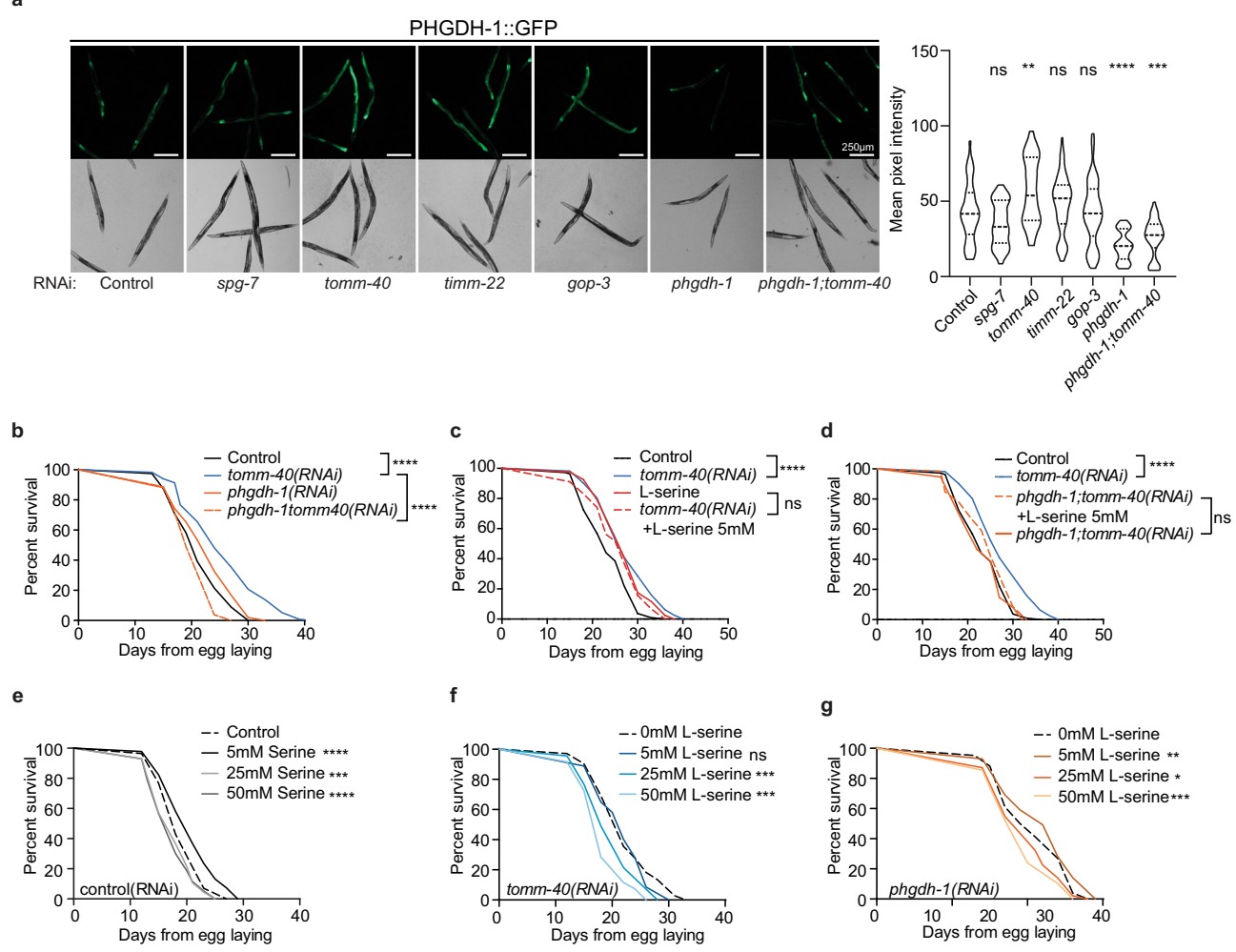

**Fig. 6 De novo serine biosynthesis is required for longevity upon MitoMISS. a** Epifluorescence images showing the total protein levels of PHGDH-1 upon *spg-7, tomm-40, timm-22, gop-3, phgdh-1,* and *phgdh-1;tomm-40* genetic inhibition (left panel) and the corresponding fluorescence quantification depicted with violin plots (right panel). $n = 3$ biologically independent experiments with at least 20 worms per condition one-way ANOVA with Tukey's multiple comparison test. Exact sample size and *P* values are included in the Source Data file. (**** denotes $P < 0.0001$, *** denotes $P < 0.001$, ** denotes $P < 0.01$ and * denotes $P < 0.05$). a.u. arbitrary units. **b** Lifespan analysis of MitoMISS animals upon inhibition of de novo serine biosynthesis. **c** Lifespan analysis of MitoMISS animals with or without of L-serine supplementation. **d** L-serine supplementation does not rescue the inhibition of de novo serine biosynthesis upon MitoMISS. **e–g** Lifespan analysis of wild type animals (**e**), MitoMISS animals (**f**) and animals with inhibited de novo serine biosynthesis (**g**) upon different concentrations of exogenous serine supplementation. $n = 2$ biologically independent experiments. Curves were compared with the Log-rank (Mantel–Cox) test (*** denotes $P < 0.0001$, ** denotes $P < 0.001$, * denotes $P < 0.01$, ns denotes not significant); detailed values are shown in Supplementary Table 2.

control or MitoMISS (Supplementary Fig. 9f). We noted that MitoMISS potently extends the lifespan of *sams-1*-mutant animals. Therefore, we conclude that MitoMiss utilizes a distinct metabolic pathway that depends on de novo serine biosynthesis to exert its effects on longevity while it acts independently of dietary serine.

**Metabolic rewiring upon MitoMISS ameliorates glucose-induced toxicity.** Dietary glucose has been linked to the limitation of lifespan in nematodes, flies, and mammals[61–64]. Conversely, restriction of dietary sugar or/and decreased flux through glycolysis has been linked to lifespan extension, improved glucose tolerance, and reduced oxidative damage in several species[65,66]. However, in our experimental setting, it became apparent that glucose incorporation, glycolysis, and rewiring of glucose metabolism towards de novo serine biosynthesis, are causatively linked to the longevity phenotype. This model predicts that MitoMISS could protect animals from dietary glucose-induced toxicity. To

test this hypothesis, we performed survival analysis of the wild type and TOMM-40-depleted animals, in the presence or absence of D-glucose. In agreement with previous studies, growing of wild type animals in the presence of D-glucose is detrimental for lifespan (Fig. 7a)[66]. Despite exhibiting increased glucose levels, MitoMISS-treated animals are partially protected from glucose-induced toxicity. Our findings suggest that MitoMISS-associated glucose channeling through de novo serine biosynthesis and the trehalose-based sugar storage, ameliorates the adverse effects of increased glucose levels.

To gain further insight into the physiology of this phenomenon, and given the disproportionate amounts of energy needed among tissues, we asked whether MitoMISS-associated longevity acts in a tissue-specific manner. Using transgenic *C. elegans* strains engineered for tissue-specific knockdown, we performed *tomm-40* RNAi specifically in neurons, hypodermis, intestine, and body-wall muscles (Fig. 7b–e)[67–69]. We noticed that the tissues that mediate the longevity effect of MitoMISS are the

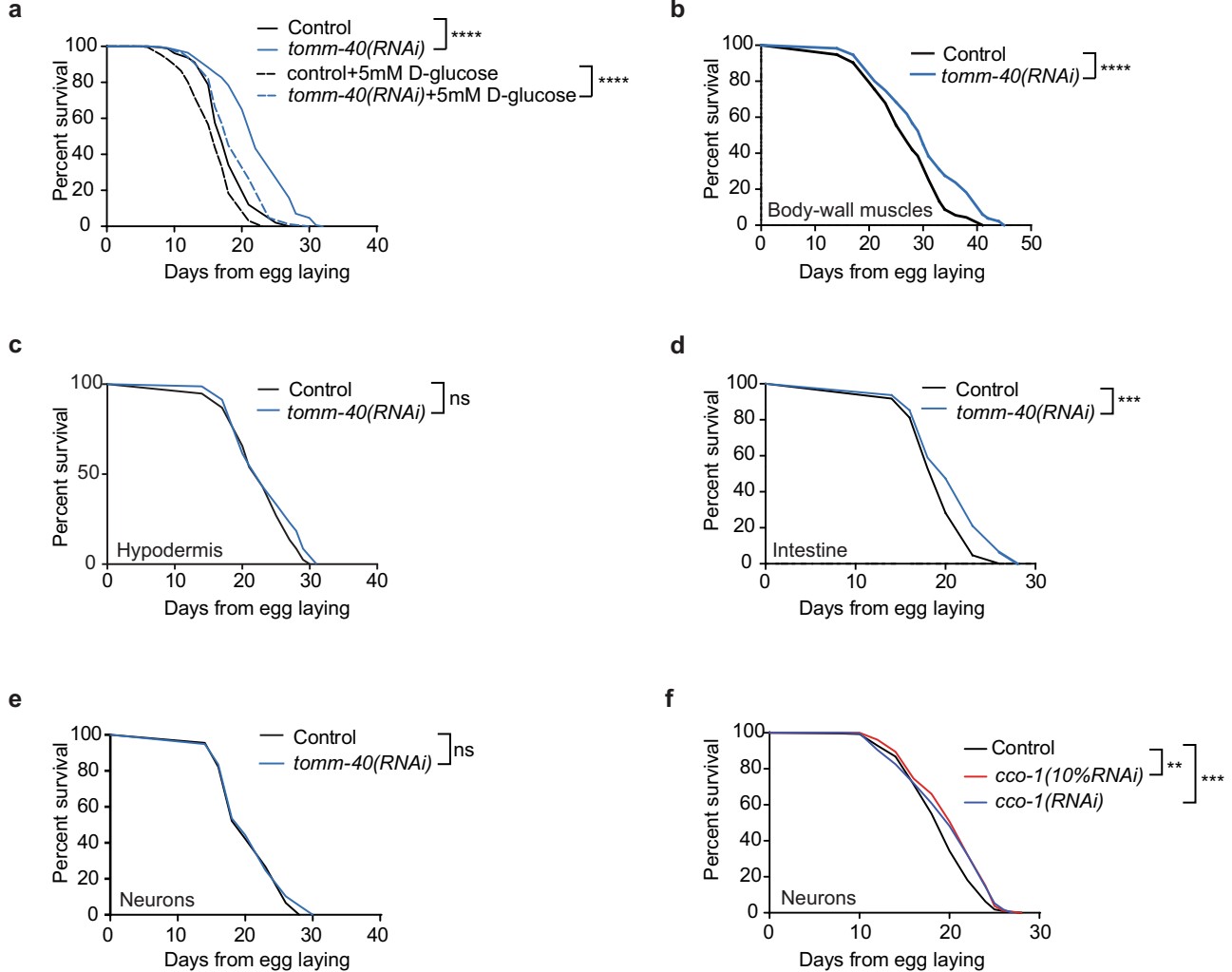

**Fig. 7 MitoMISS ameliorates glucose toxicity with regard to organismal lifespan. a** Lifespan analysis of wild type animals upon MitoMISS with and without 5mM D-glucose supplementation. **b–e** Lifespan analysis of tissue-specific MitoMISS application. **f** Neuron-specific genetic inhibition of *cco-1* 100% and 10% diluted. Survival curves were compared with the Log-rank (Mantel–Cox) test (*** denotes $P < 0.0001$, ** denotes $P < 0.001$); detailed values are shown in Supplementary Table 2.

intestine and body-wall muscles, the insulin-target tissues in the nematode (Fig. 7b, d). On the contrary, targeted MitoMISS in the hypodermis or neurons failed to extend lifespan (Fig. 7c, e). This property of *tomm-40* inhibition contrasts the reported role of neurons in mediating the longevity signal upon neuron-specific *cco-1* RNAi[48,70]. To test whether neuronal-specific *cco-1* inhibition extends lifespan in our experimental setting, we performed neuronal-specific inhibition of *cco-1* in the transgenic animals we used for tissue-specific knock-down (Fig. 7f). As previously reported, neuronal-specific silencing of *cco-1* RNAi extends lifespan, in contrast to neuronal MitoMISS. Taken together, these findings corroborate our notion that MitoMISS acts synergistically with mild mitochondrial dysfunction to extend lifespan.

## Discussion

Age-dependent decline in mitochondrial function has been linked to a multitude of pathological conditions. Therefore, improvement of mitochondrial function during aging and upon disease has been a long-standing scientific goal. Here, we identify MitoMISS as a distinct longevity pathway, unexpectedly linked to anabolic remodeling that ameliorates glucose toxicity. MitoMISS activates some features of UPR^mt which is regulated by ATFS-1.

ATFS-1 is known to be required for longevity upon mild mitochondrial dysfunction[48,49]. Moreover, recent studies describe a wide range of processes that are regulated by UPR^mt[44]. In this study, the pro-longevity function of ATFS-1 is being attributed to its role in metabolic reprogramming rather than its role in transcribing mitochondrially targeted chaperones and proteases to maintain organellar homeostasis. Notably, recent findings, implicate ATFS-1 in the expansion of the mitochondrial network during *C. elegans* development. Specifically, it was proposed that ATFS-1 is excluded from mitochondria during development as its import is antagonized by the highly expressed mitochondrial precursors, with strong mitochondrial targeting signals, that are massively synthesized during development. Therefore, ATFS-1 accumulates in the nucleus triggering the expression of several mitochondrial components, participating in mitochondrial network expansion[71]. These intriguing findings provide a putative underlying cause for the differences we monitored between MitoMISS- and *spg-7* RNAi-driven UPR^mt, as the ATFS-1-dependent expansion of the mitochondrial network is inhibited upon MitoMISS.

Mechanistically, the beneficial effects of reduced mitochondrial abundance are mediated by an anabolic shift whereby accumulated glycolytic intermediates feed into branching biosynthetic

pathways. So far, catabolic reactions induced upon starvation, caloric restriction, or nutrient shortage, have been linked to longevity in several species. Catabolic reactions like autophagy, lipid oxidation, and oxidative phosphorylation are believed to support energy production invested in repair and homeostatic mechanisms, thereby promoting lifespan and healthspan[72,73]. On the contrary, dividing cells need to acquire an adequate amount of biomass to support the production of new cells[74,75]. This requires a great deal of anabolic reactions to take place, so that new DNA and RNA molecules, new proteins, lipids, and carbohydrates are formed. Serine feeds into the folate cycle to support several anabolic reactions supporting cell proliferation of mammalian cells[75–77]. In this study, we report that an anabolic shift is causatively linked to longevity. De novo serine biosynthesis from the glycolytic intermediate glycerate-3P is central to this shift and is required for longevity under these conditions.

Furthermore, our analysis indicates increased glycolytic synthesis of glycerol and glycerol-3-phosphate suggesting an increased activity of the GL/FFA cycle, which can contribute to lipid detoxification and production of crucial metabolic intermediates to maintain cellular homeostasis. Mobilization of lipid stores upon MitoMISS needs further investigation for its role in longevity. Moreover, metabolomic analysis confirmed increased cycling between glucose and trehalose, a condition that has been previously reported to correlate with longevity[56,78].

Dietary supplementation of several amino acids, namely methionine, serine, glycine, histidine, arginine, and lysine, has been shown to promote longevity in nematodes[59]. Serine and glycine are non-essential amino acids which, apart from their role in protein synthesis, participate in crucial metabolic pathways for the cellular function[79]. Serine and glycine metabolism have been associated with cell proliferation in cancer cells, which exhibit mitochondrial dysfunction[75,80,81]. Moreover, supplementation of extracellular glycine has been considered beneficial in obesity and metabolic diseases[82]. Treatment of human cancer cells with several mitochondrial poisons leads to ATF4-dependent increase in de novo serine biosynthesis[83]. Recently, glycine supplementation was shown to extend lifespan also in female and male mice[84]. We found that MitoMISS triggers an increase of intracellular levels of serine, glycine, and threonine. This increase could be due either to increased serine uptake or to increased de novo serine biosynthesis. Notably, we found that inhibition of de novo biosynthesis abolishes MitoMISS-associated longevity and propose that the de novo branch is causatively linked to the longevity phenotype. Interestingly, inhibition of de novo serine biosynthesis cannot be rescued by exogenous serine supplementation. The same paradox exists in several types of cancer cells where inhibition of PHGDH and de novo serine biosynthesis when environmental serine is abundant impairs cancer cell proliferation[75,85,86]. Several hypotheses were put forward to unravel this conundrum. Recently, it was proposed that PHGDH inhibition causes alteration in nucleotide levels independently of serine utilization[87]. Our study provides a framework for elucidating the exact mechanism whereby PHGDH contributes to longevity, and paves the way for designing tissue-specific interventions, aiming to alleviate metabolic disease-associated pathology.

## Methods

**Strains and genetics**. We followed standard procedures for *C. elegans* strain maintenance[88]. Nematode rearing temperature was kept at 20 °C, unless noted otherwise. The following strains were obtained from *Caenorhabditis* Genetics Center (Minneapolis, MN): N2: wild type Bristol isolate, SJ4100 *zcIs13* [*hsp-6p*::GFP] V, SJ4058 *zcIs9* [*hsp-60*::GFP] V, SJ4197 *zcIs39* [dve-1p::dve-1::GFP] II, SJ4151 *zcIs19* [ubl-5p::ubl-5::GFP], CF1038 *daf-16(mu86)* I, CB1370 *daf-2(e1370)* III, DA453 *eat-2(ad453)* II, EU1 *skn-1(zn67)*, EU40 *skn-1(zu129)*, EU31 *skn-1(zu135)*, RB754 *aak-2(ok524)* X, OH149 *ceh-23(ms23)* III, XY1054 *cep-1(lg12501)* I, VC172 *cep-1(gk138)* I, ZG31 *hif-1(ia4)* V, BC10307 *dpy-5(e907)* I; sEx10307, TU3401 *sid-1(pk3321)* V; uIs69 V, VP303 *rde-1(ne219)* V; kbIs7 [nhx-2p::rde-1+rol-6(su1006)], NR222 *rde-1(ne219)* V; kzIs9 [pKK1260(lin-26p::nls::gfp), pKK1253(lin-26p::rde-1), pRF4(rol-6 marker)], NR350 *rde-1(ne219)* V; kzIs20 [pKK1253(lin-26p::rde-1), pTG96(sur-5p::nls::gfp)], SJ4197 *zcIs39*[p_dve-1p::DVE-1::GFP] II, SJ4151 *zcIs19* [ubl-5p::ubl-5::GFP], RB867 *haf-1(ok705)* IV, JJ2586 cox-4(zu476[cox-4::eGFP::3xFLAG]) I, RW12254 *hsf-1*(st12254[hsf-1::TY1::EGFP::3x-FLAG]) I, CL2070 *dvIs70* [*hsp-16.2p*::GFP+*rol-6(su1006)*], SJ4005 *zcIs4* [*hsp-4*::GFP] V, CF2893 sEx1112δ [gpd-2p::GFP+(pCeh361)dpy-5(+)], EU2917 drp-1(or1941[drp-1::GFP]) IV, BC10307 *dpy-5(e907)* I; sEx10307 [rCes Y87G2A.8::GFP+pCeh361], CF1553 *muIs84* [(pAD76) *sod-3p*::GFP+*rol-6(su1006)*], BC14279 *dpy-5(e907)*I; sEx14279 [rCesF42G8.12::GFP+pCeh361], CL2166 *dvIs19* [(pAF15) *gst-4p*::GFP::NLS] III, BC11314 *dpy-5(e907)* I; sEx11314 [rCesT20G5.2::GFP +pCeh361], BC10551 *dpy-5(e907)* I; sEx10551 [rCesF20H11.3::GFP+pCeh361], N2; *Ex*[P_CEOP1740::GFP, pRF4]

The following strains were obtained from the National BioResource Project (Tokyo, Japan): *atfs-1(tm4525)* V, *dve-1(tm4803)* X, *fgt-1(tm3165)* II. Strains produced during this study by crosses: *atfs-1(tm4525)*;zcIs9 [*hsp-60*::GFP] V, *haf-1(ok705)* IV;zcIs9 [*hsp-60*::GFP] V, *dve-1(tm4803)* X; zcIs9 [*hsp-60*::GFP] V. Strains produced in this study by micro-injections: N2;Ex001[p_fgt-1 GFP;pRF4], N2;Ex001[p_phgdh-1 PHGDH-1::GFP;pRF4] and microparticle bombardment: *unc-119(ed3); Ex*[P_myo-2 Glifon4000; *unc-119(+)*].

**Feeding RNAi**. Feeding RNAi was performed as previously described[89]. All RNAi treatments were started upon hatching and proceeded throughout post-embryonic development and adulthood. Simultaneous RNAi knockdown of two genes was achieved either by cloning both genes in the same vector (*phgdh-40;tomm-40, gpi-1;tomm-40, atfs-1;tomm-40, dve-1;tomm-40*) within the two T7 RNA polymerase promoters, or by combining different dsRNA expressing bacteria. For *cco-1/timm-23* and *atp-3/timm-23* double RNAi, we combined *cco-1* or *atp-3* RNAi bacteria with *timm-23* RNAi bacteria in a 1:9 ratio (1/10 dilution for *cco-1* or *atp-3* and 9/10 dilution for *timm-23*). This dilution has been shown to have maximum lifespan prolonging capacity for the respective genes according to previously published results[90]. For *ucr-1-timm-23* double RNAi, we combined *timm-23* and *ucr-1* RNAi bacteria in a 1:1 ratio (1/2 dilution for each gene). The single RNAi treatments in these experiments were achieved by similar dilutions with the empty vector (pL4440) bearing bacteria (control RNAi).

**Molecular cloning**. Cloning strategy for all RNAi constructs entails PCR amplification of gene of interest from the nematode genomic DNA or cDNA followed by cloning of each PCR fragment into pCR-II TOPO vector (Invitrogen). cDNA was synthesized using the PrimeScript Reverse Transferase kit (Takara). Proper restriction enzymes were used to subclone genes from pCR-II TOPO vector into pL4440 vector for T7 dependent expression of dsRNA. For *atfs-1* RNAi construct, a PCR fragment of 3013 bp using cDNA as template was cloned in pCR-II TOPO vector and after digestion with *Kpn*I/*Xba*I was inserted into pL4440 vector. Double RNAi vector of *atfs-1/tomm-40* was generated by subcloning *atfs-1* digestion fragment using *Kpn*I/*Xba*I restriction enzymes into *tomm-40* containing pL4440 vector. To generate *gpi-1* and *phgdh-1*(C13C9.2.1) RNAi constructs, PCR fragments of 2824 and 810 bp, respectively, were initially inserted into pCR-II TOPO vector and after digestion with *Kpn*I/*Apa*I were inserted into pL4440 vector. For double RNAi constructs of *gpi-1;tomm-40* and *phgdh-1*(C13C9.2.1)/*tomm-40*, digestion fragments using *Kpn*I/*Apa*I were inserted into *tomm-40* containing pL4440 vector. For *dve-1* RNAi construct, a PCR fragment of 1507 bp using cDNA as template was cloned into pCR-II TOPO vector and after digestion with *Xba*I/*Spe*I was inserted into pL4440 vector. Double RNAi construct of *dve-1;tomm-40* was generated by excision of *dve-1* fragment from pCR-II TOPO using *Kpn*I/*Apa*I restriction enzymes and subsequently ligated into *tomm-40* containing pL4440.For hsp-60 RNAi construct, a PCR fragment of 2354 bp was initially inserted into pCR-II TOPO vector and subsequently cloned into pL4440 using *Eco*RI restriction enzyme. Green Glifon4000 was a gift from Tetsuya Kitaguchi (Addgene plasmid # 126208; RRID: Addgene_126208)[54]. PCR amplified Glifon4000 was cloned into pPD95.77 after removal of GFP using *Bam*HI/*Eco*RI restriction sites. For the construction of *fgt-1* transcriptional reporter line, we PCR amplified the promoter region of *fgt-1* and subcloned it in pPD95.77 vector upstream of GFP coding region. Subsequently, pPD95.77 vector was injected in wild-type worms together with the co-transformation marker, pRF4 rol-6(su1006). N2;Ex001[p_fgt-1GFP;pRF4] line was maintained by selecting rollers. The primers used for each gene as well as the target vector are provided in (Supplementary Table 1).

**Protein analysis and antibodies**. Synchronous animal populations under all conditions tested were harvested and used for whole-worm lysates preparation. For total worm protein extraction, protein samples were produced by directly boiling worms in 23 Laemmli sample buffer with b-mercaptoethanol supplemented with 5 mM PMSF and a complete mini proteinase inhibitor cocktail (Roche). Protein samples were analyzed by 8% Tricine-SDS–polyacrylamide gel electrophoresis (SDS–PAGE), transferred on nitrocellulose membrane, and blotted against various antibodies. Particularly, the antibodies used for this study were anti-alpha-tubulin (DSHB 12G10) used in dilution 1/5000, anti-atp5A (Abcam, ab14748), used in dilution 1/1000, anti-eiF2 subunit 1α (Cell Signaling, 9722), used in dilution 1/1000 and anti-phospho-eiF2α (Cell Signaling, 119A11), used in dilution 1/1000.

**RNA isolation and qRT-PCR analysis**. Total mRNA was isolated from synchronized day 1 adults which were lysed in 250 µl of TRIzol by freeze-cracking (Invitrogen). For cDNA synthesis, mRNA was reverse transcribed using iScriptTM cDNA Synthesis Kit (BioRad). Quantitative PCR was performed in triplicate using a BioRad CFX96 Real-Time PCR system (BioRad). Expression levels of the housekeeping genes *act-3* and/or *pmp-3* were used as an internal control for normalization. Unless stated otherwise, at least three independent experiments were performed for each condition. For mtDNA quantification by qRT-PCR we used primers hybridizing on mtDNA or nDNA encoded genes (*atp-6* and *ama-1* respectively). All qPCR primer sequences used are included in Supplementary Table 1.

**Lifespan assays**. Lifespan assays were performed at 20 °C. Synchronous animal populations were generated by hypochlorite treatment of gravid adults to obtain tightly synchronized embryos that were allowed to develop into adulthood. The day of egg harvest and initiation of RNAi was used as day 1. All RNAi treatments started from hatching. Sterilized water solution of dextrose was added in NGM medium prior to plate pouring to a final concentration of 5 mM. Sterilized water solution of serine was added on the top of the RNAi bacterial lawn. The final serine concentration was estimated to be 5, 25, or 50 mM, based on the volume of the NGM medium per plate. Animals were transferred to fresh plates every 2–4 days thereafter and examined every other day for touch-provoked movement and pharyngeal pumping, until death. Worms that died due to internally hatched eggs, extruded gonad or desiccation due to crawling on the edge of the plates, were censored off the experiment. Median lifespans and statistical significance of compared survival curves are provided in Supplementary Table 2.

**Glucose uptake assay**. To assess glucose uptake in live animals, we modified a previously established glucose tracing assay using 2-NBDG fluorescently-tagged deoxyglucose analog[66]. Briefly, animals were grown on regular RNAi plates for ~2 days until the late L4 developmental stage. Then, control and RNAi-treated individual animals were picked and transferred into 24-well plates containing UV-killed bacteria grown on 500 µl medium per well. Each well contained a final concentration of 0.15 mM of 2-NBDG. After 24 h feeding with 2-NBDG, animals were mounted and paralyzed with levamisole on standard microscope slides prior to microscopic observation.

**Imaging**. Transgenic animals carrying the indicated somatic fluorescence reporters were synchronized and grown to the desired age. For live imaging, worms were mounted and paralyzed with levamisole on standard microscope slides. Microscopic examination of worms was performed with a ZEIS AxioImager Z2 epifluorescence or a ZEIS LSM 710 confocal microscope. We calculated the mean and maximum pixel intensity for each animal in these images using the ImageJ software (http://rsb.info.nih.gov/ij/).

**ATP measurements**. To determine ATP content, 50 1-day-old adult hermaphrodites were collected in M9 buffer, washed twice, and frozen at −80 °C. Frozen worms were immersed in boiling water for 15 min, cooled and centrifuged to pellet insoluble debris. The supernatant was moved to a fresh tube and diluted ten-fold before measurement. ATP content was determined by using the Roche ATP bioluminescent assay kit HSII (Roche Applied Science) and a TD-20/20 luminometer (Turner Designs). ATP levels were normalized to total protein content. We used the Prism software package (GraphPad Software) to carry out statistical analyses.

**Sample collection for metabolomic analysis**. Wild type worms were grown in *ad libitum* conditions for three generations. Gravid adults were incubated with hypochoride solution until their bodies were disassembled and the eggs were isolated. Eggs were placed on 10 cm Petri dishes that contain medium for RNAi induction and were seeded with the respective bacterial strain. Worms were grown on these plates from hatching until young adults, for 2.5 days at 20 °C. Then they were pooled, washed twice with ice-cold 0.9% (w/v) high-purity sodium chloride (NaCl—Honeywell Riedel-de Haën®, Germany) in HPLC-grade water (Panreac, Spain) with ddH2O and pelleted. Worm pellets were flash-frozen in liquid nitrogen, weighted, and stored until metabolite extraction. Most worm pellet samples weighted ~100 mg. If not, appropriate normalization in the metabolic profiles was carried out. The samples were stored at −80 °C until analysis and when transport was needed between laboratories this was carried out in dry ice.

**Untargeted GC–MS metabolomics: data acquisition & normalization**. The polar metabolite profiles of the wild type animals treated with control or *tomm-40* RNAi were acquired and analyzed at the Metabolic Engineering & Systems Biology Laboratory (MESBL) of FORTH/ICE-HT using untargeted gas chromatography–mass spectrometry (GC–MS) metabolomics. Extraction and profile acquisition protocols described in[91] were appropriately adapted to *C. elegans* specifics. Each frozen worm pellet sample had to be initially ground to powder form in a porcelain mortar with the addition of liquid nitrogen. Each sample was collected from the mortar with ice-cold methanol in a 2 ml Eppendorf tube, into which

1 µg ribitol (Alfa Aesar, Germany) and 2 µg [U-¹³C]-glucose (Cambridge Isotope Laboratories, USA) were added as internal standards. The samples underwent the extraction procedure described in refs. [92,93]. After centrifugation, supernatants from worm pellets weighing more than 60 mg were split into two (a, b) equal parts; the rest were not divided. The dried extract of each part was then derivatized to its (MeOx)TMS-derivatives through reaction with 50 µl of 20 mg/ml methoxyamine hydrochloride (Alfa Aesar, Germany), in pyridine (Carlo Erba Reagents, Italy) for 90 min, followed by reaction with 100 µl N-methyl-trimethylsilyl-trifluoroacetamide (MSTFA) (Alfa Aesar, Germany) at 40 °C for at least 6h[93,94]. The metabolic profile of at least one part (a and/or b) of each sample was measured at least thrice at different derivatization times. The peak identification and quantification were carried out using the MS Workstation software v.6.4 based on the commercial NIST (v.2.0 as implemented in the MS Workstation software) and our in-house MESBL peak library[91,94]. The metabolic profile data validation, normalization, and filtering were carried out using the M-IOLITE software suite (http://miolite2.iceht.forth.gr)[94], estimating the Relative Peak Areas (RPAs) of the marker ions of each metabolite derivative with respect to the peak area of the internal standard ribitol ion 319. After data correction, normalization and filtering, the normalized profiles comprised 72 metabolites, consistently detected in all samples at both collection times (spermidine was detected only in the samples of the first collection time). The metabolic profile of each sample or sample part (a or b), if divided at the extraction stage, was estimated as the mean of the normalized profiles of all its technical replicates. The raw metabolomic data set as identified and quantified from the MS-reconstructed chromatograms using MS Workstation software v. 6.4 is provided in Supplementary Data File 1A. The .sms files of the MS-reconstructed chromatograms can be accessed via a link provided in Supplementary Data File 1A. The final normalized metabolic data set used in the further analysis is provided in Supplementary Data File 1B with all profiles normalized to correspond to an original worm pellet weight of 100 mg. Supplementary Data File 1C shows the % fraction of each metabolite in the total RPA of each profile.

**Metabolomic data set multivariate statistical analysis**. The untargeted GC–MS metabolomic datasets acquired in this study were analyzed as described in the methodological paper of our group[91]. More specifically, hierarchical clustering (HCL) and significance analysis of microarrays (SAM) algorithms were applied as implemented in the open-source TM4 MultiExperiment Viewer (MeV) software (version 4.9.0)[95], for the profiles of each collection time (CT1 and CT2). The SAM method has been designed for the analysis of omic data, as they are not expected to follow a particular distribution, as is the case for the student's *t*-test or the F-test, and are inter-dependent through their connectivity in the metabolic, in particular, and the larger biomolecular network, in general, of the examined biological system. The SAM method identifies metabolites with significantly higher or lower abundance in a metabolic profile group, considered as Group B compared to a metabolic profile group, considered as Group A, for a significance threshold delta "δ" corresponding to a particular false discovery rate (FDR)—median. The metabolites with significantly higher or lower abundance in Group B vs Group A are referred to as "positively" or "negatively" significant metabolites, respectively, of the particular comparison, respectively. All described SAM analyses were performed using the smallest significance threshold delta (δ) for which the false discovery rate (FDR)—median was zero (strictest) or the smallest possible significance threshold delta for a particular comparison[96,97].

**Metabolic network reconstruction**. The primary metabolic network of nematodes from glucose to lactate including the glycerolipid/free fatty acid (GL/FFA) cycle and the serine/glycine/threonine metabolism branch was reconstructed based on information from metabolic databases (KEGG)[98] and ExPASy[99] and the literature. The network was designed using the CorelDRAW X7 software.

**Reporting summary**. Further information on research design is available in the Nature Research Reporting Summary linked to this article.

## Data availability
Raw data from all figures and supplementary figures are included in the source data file. The raw data from the metabolomics analysis are provided as a separate sheet in Supplementary Data File 1. The respective chromatographs are accessible via this link: https://tavernarakislab.gr/publications/lionaki_et_al.zip. Source data are provided with this paper.

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

## Acknowledgements

We thank Aggela Pasparaki for technical assistance and all members of our labs for useful discussions. This project was supported by the European Research Council, under grant agreement "ERC-GA695190-MANNA" to N.T., the Hellenic Foundation for Research and Innovation (HFRI) and the General Secretariat for Research and Technology (GSRT), under Grant Agreement No. [1898] to E.L. The metabolomic and bioinformatic analyses at FORTH/ICE-HT were supported by the projects "BITAΔ: Advanced Research Activities in Biomedical and Agro alimentary Technologies" (MIS 5002469), implemented under the "Action for the Strategic Development on the Research and Technological Sector" and "ELIXIR-GR: The Greek Research Infrastructure for Data Management and Analysis in Life Sciences" (MIS 5002780), implemented under the Action "Reinforcement of the Research and Innovation Infrastructure", both funded by the Operational Program "Competitiveness, Entrepreneurship and Innovation"(NSRF 2014-2020) and co-financed by Greece and the European Union (European Regional Development Fund).

## Author contributions

E.L. and N.T. conceptualized the study. E.L., I.G. and I.D. designed the methodology, performed experiments, and analyzed the data. M.K.I. and M.I.K. performed and analyzed metabolomics experiments. E.L., I.G., M.K.I., M.I.K. and N.T. wrote the first manuscript and all the authors reviewed and edited the text. N.T., M.I.K. and E.L. provided funding for the study.

## Competing interests

The authors declare no competing interests.

## Additional information

**Peer review information** *Nature Communications* thanks Martin Denzel and the other anonymous reviewer(s) for their contribution to the peer review this work. Peer reviewer reports are available.

