## [Peer Review File · Nature Communications]

Reviewers' Comments:

Reviewer #1:

Remarks to the Author:

In the current study, Lionaki et al show that inhibition of the mitochondrial protein import system (mainly through RNAi of *tim-23* and *tom-40*), referred to as MitoMISS, extends the lifespan of *C. elegans*, an effect independent of multiple canonical longevity pathways (e.g. insulin signaling and caloric restriction). They further show that MitoMISS regulates lifespan extension likely through a mechanism linked to an adaptive metabolic shift through increasing glucose uptake, glycolysis and de novo serine biosynthesis, which is "unexpectedly" controlled by the UPRmt transcription factor ATFS-1. Moreover, they suggest that the role of MitoMISS in metabolic reprogramming and lifespan extension is uncoupled from the canonical UPRmt as they found that: 1) MitoMISS further extends lifespan of animals that experience mild mitochondrial dysfunction (e.g. *cco-1* RNAi), 2) targeted MitoMISS in the neurons failed to extend lifespan, while neuron-specific *cco-1* RNAi has been reported to function in lifespan extension.

The paper presents an interesting angle to study the impact of mitochondrial import suppression on energy homeostasis and lifespan extension, but is at best preliminary as it stands.

Unfortunately, there are several studies that document the impact of abnormal mitochondrial protein import on mitochondrial function in multiple organisms seriously limiting the novelty of the observations and contesting some of the conclusions made. Finally and most importantly, before it can be concluded that the proposed "novel" MitoMISS acts differently from the canonical UPRmt or proteotoxic stress signaling, more mechanistic analysis are required to support their key claims of the study.

Major comments

- A panoply of studies has addressed mitochondrial import abnormalities, e.g. MitoTAD, mPOS, UPRam, mitoCPR, MAGIC, seriously limiting the novelty of the observations (a non-exhaustive list of the most recent papers - PMID : 31118508, 29650645, 28241148). Most of these papers show that proteotoxic stress, not metabolic disturbances, cause the phenotypic abnormalities, which is opposite to the observations reported here. Yet these papers are not properly discussed within the context of the data generated. The "novel" MitoMISS that the authors investigate is furthermore actually already presented in a previous study from Haynes group (PMID: 22700657, also cited by the authors).
- The conclusion that MitoMISS extends the lifespan of *C. elegans* independent of the canonical UPRmt is unfortunately not convincing and somehow misleading. In my view, the MitoMISS is more likely to belong to canonical UPRmt, as evidenced with the induction of classical UPRmt reporter genes *hsp-6* and *hsp-60* as well as the ATFS-1 dependency. For the results that "MitoMISS further extends lifespan during mild mitochondrial" dysfunction (Figure 2G and 2H), the use of only 10% of *cco-1* or *atp-3* RNAi (as indicated in the methods) make me doubt that the "mild mitochondrial dysfunction-induced lifespan extension" is already "saturated". RNAi of *tim-23* could simply further exacerbate the mitochondrial dysfunction, leading to a more robust UPRmt response and thereby further extend the lifespan (the author should use qRT-PCR and/or western blots to compare the expression of UPRmt genes to clarify this point). For the data that "targeted MitoMISS in the neurons failed to extend lifespan" (Figure 6E), the methods used by the authors is obviously different from the previously paper to knockdown *cco-1* specifically in the neuron (where a transgenic worm line carrying an inverted repeat hairpin (HP) targeting *cco-1* was created) (PMID: 21215371).
- The initial rationale of the paper it is not clear. The authors decided to impact mitochondrial mass by impairing the mitochondrial protein import machinery. However, they do not give any rationale or reference to justify this approach. One could consider that a more straightforward approach to target mitochondrial mass would be to impact mitochondrial biogenesis. This could be achieved for example by genetically inhibiting TFAM homologue *hmg-5*, or DNA polymerase gamma homologue *polg-1*. Such data are required to make the observations of MitoMISS generalizable.
- The use of muscle and intestine-specific mitochondrial reporters to indicate the "mitochondrial mass" may not reflect the reality; the authors should also determine the expression of mitochondrial proteins (e.g. ATP5A, NDUFS3) with western blots during MitoMISS. Also, does MitoMISS affect the mtDNA/nucDNA ratio?
- The authors overall relied a lot on the GFP expression intensity of individual reporter strains,

they should also use qRT-PCR to verify at least some of the key results (e.g. Figures 2D, 3D, 4A, 4D).

- Why do tomm-40, timm-23, timm-22 and gop-3 RNAi's give so different results regarding ubl-5 induction.
- Why DVE-1 and UBL-5 are not required for the observed phenotypes is unclear.
- In line 220-224. The authors make a bold statement by excluding the participation of chaperones and proteases to the observed effect just performing experiments using gpi-1 RNAi. They should use RNAis against important targets of ATFS-1 to demonstrate this point.

Minor comments

- The figures panels are often referred in the text not following alphabetical (logical) order. This is confusing for the reader and makes it difficult to follow the flow of the story. The authors should properly reference the figure panels in which the data are presented each time. Moreover, the lack of figures numbers in the figure panels exacerbates this issue.
- The authors present the same data in Figure 2F and Figure S1G (in the latter they also include other conditions). It is not standard to show the same lifespan curves in two different figures.
- In Figure 2G it would be interesting to know whether cco-1(RNAi);timm23(RNAi) is statistically different from cco-1(RNAi) alone. Same goes for Figure 2H, timm23(RNAi);atp-3(RNAi) and atp-3(RNAi)

Reviewer #2:

Remarks to the Author:

The manuscript by Lionaki et al investigates the role of mitochondrial mass in *C. elegans* longevity. It addresses important questions that remain unresolved in the context of mitochondrial longevity and the role of the UPR-mt in lifespan extension. The paper shows that RNAi-mediated depletion of the ribosomal import pores tomm-40 and timm-23 depletes mitochondrial content, ROS, and ATP. Knockdown of these genes, but not of other mitochondrial import factors such as timm-22, extends lifespan. This appears to be independent of insulin signalling or lifespan extension mediated by SKN-1. Inhibition of mitochondrial import triggers the atfs-1 dependent induction of hsp-60 and hsp-6 and the longevity is, likewise, atfs-1 dependent. Glycolysis genes were affected by tomm-40 and timm-23 RNAi and this was mirrored by enhanced glucose uptake. Metabolome analysis then confirmed changes in glycolysis and revealed increased steady state levels of serine. Of note, serine supplementation did not further extend the survival of tomm-40 RNAi treated worms. Finally, the authors use tissue specific RNAi and find that knockdown in the body wall muscle and intestine are sufficient for lifespan extension.

The manuscript is concisely and clearly written and documents a number of very interesting observations that stand to enhance the knowledge in the field of mitochondrial longevity and metabolism. However, some of the conclusions are not solidly supported by the data and their interpretation is at times unclear.

Major points:

- 1) A general characterization of the MitoMISS animals would add important insights. Given the low ATP levels, what is the effect on brood size, developmental time and all-over morphology during the knockdown?
- 2) It would be important to solidify the MitoMISS phenotype regarding mitochondria. Is there a reduction of mito DNA copy number? Are there changes in biochemical markers of mitochondrial abundance?
- 3) Given that the MitoMISS phenotype is based on RNAi, it would be important to quantify the efficiency of the knockdown of the 4 genes in Figure 1 using qPCR.
- 4) Is there an induction of GCN-2 in the MitoMISS animals (See Baker et al., 2012)? A GCN-2/eIF2alpha induction might be expected in animals with reduced mitochondria and atfs-1 induction.

5) Generally, in many lifespan assays, the untreated/wild type controls were not included. This is generally acceptable but in Fig 3E, given the absence of an effect, it would be important to establish the effect of the RNAi treatment in the same experiment.

6) Fig 4: more experiments are needed to test if really an increased glucose flux is linked to lifespan extension in the MitoMISS animals. A transcriptional reporter for *gpi-1* is not sufficient to conclude that there is increased glycolysis (line 212). The fact that *gpi-1* RNAi alone extends lifespan makes it hard to interpret the data as, in a scenario where *gpi-1* is required for MitoMISS longevity, the survival of the *gpi-1;tomm-40* double treatment should phenocopy the *gpi-1* single-treatment. Here qPCR is needed to check the effect of the double vs single RNAi treatments.

7) In a similar argument, *fgt-1* is induced by *tomm-40* RNAi, but is this responsible for the increased glucose uptake (as done in Fig 4C?) In Fig 4D, better fluorescence images are needed, along with DIC images. Can the increase of *fgt-1* expression be confirmed by qPCR?

8) Increased lactate and glucose concentrations in Sup Table 2 can be used as an argument for more glycolytic flux. Perhaps it's worth considering changing Figs 4 and 5 to better support the idea of increased glycolysis.

9) Fig 5C: the authors observe increased serine levels in MitoMISS worms and test if this is required for longevity by suppressing expression of PHGDH. This RNAi treatment by itself extends lifespan. However, *phgdh-1;tomm-40* double knockdown results in WT lifespan. To interpret these data, it would be important to measure serine levels in each of the conditions. Does serine elevation or reduction extend lifespan? Or possibly both? (Does serine supplementation suppress the *phgdh-1* RNAi lifespan extension?)

10) Fig 6A: the authors state that MitoMISS worms are rescued from glucose toxicity (line 335). I don't fully agree with this interpretation as glucose is still toxic to *tomm-40* RNAi treated worms. *tomm-30* RNAi still extends lifespan under glucose, but glucose remains toxic. To get a good idea of this effect it would be important to quantify glucose in the various experimental conditions of Fig 6A.

11) The longevity of ETC inhibition shows a unique temporal requirement for lifespan extension within the UPR-mt. At what age is MitoMISS RNAi treatment required for lifespan extension? While I see that this might be beyond the scope it would be a pity not to address such an exciting question in the manuscript.

Minor points:

1) It would be great to capture better images for Fig. 4C, including DIC images. The signal looks very much like it would be associated with the gut lumen, which is concerning despite the elegant controls with D-Glucose. Please define dex in the figure legend.

2) Fig. 5A is not readable and there is generally not enough information for the reader in the figure. Are the changes in metabolite levels significant? What is the meaning of the coloured boxes in Fig 5B?

3) Figures S2C and D are presented without a positive control. It would be important to add one to show the induced state of the assay.

Manuscript :

1) For some of the Figures, the data are organized in a different order compared to the text (e.g. Fig 1C and D). Please mention GST-4 in the results text on page 6. UBL-5 and HAF-1 are not

introduced before they are referenced on page 8.

2) There are two Supplementary Table 2 files

3) In the Supplementary Table with the lifespan data it would be good to show which experiments were done as a group at the same time. Please include in the figure legend the number of repeats for each lifespan.

4) Nomenclature of molecules should be the same in text and figures (glycerate-3P vs 3-phosphoglycerate, line 291 and Fig5B).

February 15, 2021

Ref: *Nature Communications* ms: NCOMMS-20-31240-T

Dear Referees,

We would like to thank you for your time and effort in reviewing our manuscript. We considered each of the points in the reviews very carefully and made every possible effort to address them experimentally. In doing so, we strived to include substantial additional information and data, both within the main text/figures, and also in the Supplementary Information section. We believe that with your encouraging and constructive input, we have been able to resubmit a significantly more rigid and comprehensive report.

Our paper now includes a total of **13** figures (**7** main and **6** supplementary figures), comprising **102** panels in total, as well as, **3** extended supplementary tables. A point-by-point response to all the comments follows below (original comments are quoted in **bold**).

Referee 1

In the current study, Lionaki et al show that inhibition of the mitochondrial protein import system (mainly through RNAi of *timm-23* and *tomm-40*), referred to as MitoMISS, extends the lifespan of *C. elegans*, an effect independent of multiple canonical longevity pathways (e.g. insulin signaling and caloric restriction). They further show that MitoMISS regulates lifespan extension likely through a mechanism linked to an adaptive metabolic shift through increasing glucose uptake, glycolysis and de novo serine biosynthesis, which is “unexpectedly” controlled by the UPRmt transcription factor ATFS-1. Moreover, they suggest that the role of MitoMISS in metabolic reprogramming and lifespan extension is uncoupled from the canonical UPRmt as they found that: 1) MitoMISS further extends lifespan of animals that experience mild mitochondrial dysfunction (e. g. *cco-1* RNAi), 2) targeted MitoMISS in the neurons failed to extend lifespan, while neuron-specific *cco-1* RNAi has been reported to function in lifespan extension. The paper presents an interesting angle to study the impact of mitochondrial import suppression on energy homeostasis and lifespan extension, but is at best preliminary as it stands. Unfortunately, there are several studies that document the impact of abnormal

mitochondrial protein import on mitochondrial function in multiple organism seriously limiting the novelty of the observations and contesting some of the conclusions made. Finally and most importantly, before it can be concluded that the proposed "novel" MitoMISS acts differently from the canonical UPR^{mt} or proteotoxic stress signaling, more mechanistic analysis are required to support they key claims of the study.

We thank the Referee for acknowledging the significance of our work. Based on their constructive comments, we have now added a significant amount of new data corroborating our initial findings.

Major comments

A panoply of studies has addressed mitochondrial import abnormalities, e.g. MitoTAD, mPOS, UPR^{am}, mitoCPR, MAGIC, seriously limiting the novelty of the observations (a non-exhaustive list of the most recent papers - PMID: 31118508, 29650645, 28241148). Most of these papers show that proteotoxic stress, not metabolic disturbances, cause the phenotypic abnormalities, which is opposite to the observations reported here. Yet these papers are not properly discussed within the context of the data generated. The "novel" MitoMISS that the authors investigate is furthermore actually already presented in a previous study from Haynes group (PMID: 22700657, also cited by the authors).

The point raised by the Referee, regarding how proteotoxicity in the cytoplasm is involved in MitoMISS related longevity, is important, and it was one of our initial hypotheses, that we tested experimentally. In the revised manuscript we discuss the published work on the proteotoxic effects of mitochondrial dysfunction (please see introduction). Moreover, we have added a new supplementary figure (Supplementary figure 3), in which we show that neither the unfolded protein response in the endoplasmic reticulum (UPR^{ER}) nor the heat shock response (HSR) in the cytoplasm are induced upon MitoMISS or any mitochondrial protein import perturbation. To address these questions we used the transcriptional reporters of *hsp-4* and *hsp-16.2* genes (for UPR^{ER} and HSR respectively) and the translational reporter of HSF-1 (for HSR). These data suggest that a robust proteotoxic response outside mitochondria is not detected, something that has been previously shown (PMID: 27610574). Furthermore, the ratio of phosphorylated eIF2alpha to total eIF2alpha is significantly decreased upon all mitochondrial protein import perturbations (Supplementary Figure 3d), corroborating the notion that proteotoxic stress is not the driving mechanism in our case. Nevertheless since, ATFS-1, the transcription factor governing UPR^{mt} in *C. elegans*, has been shown to induce expression of proteasomal genes, we cannot exclude the possibility that, at least to some extent, a proteotoxic response is induced upon mitochondrial protein import inhibition (PMID: 25773600). Notably, since we were not able to detect significant evidence of induced proteotoxicity upon mitochondrial protein import inhibition, by any means used (as shown in Supplementary Figure 3), we believe that this is highly unlikely to be causatively linked to the observed longevity. Instead, we found a metabolic rewiring which involves *de novo* serine biosynthesis to be causatively linked to MitoMISS-associated longevity. However, we don't infer that this is the sole mechanism of action.

In the revised manuscript we have added new data corroborating the induction of *de novo* serine biosynthesis. Specifically, we have constructed a translational reporter line which expresses full length *phgdh-1* fused with GFP under its endogenous promoter. We show that MitoMISS induces an increase in PHGDH-1 protein levels in contrast to other mitochondrial protein import perturbations (Figure 6i). This finding strengthens our model that MitoMISS leads to an adaptive metabolic shift towards *de novo* serine biosynthesis.

We agree with the Referee that mitochondrial protein import and how it affects cellular physiology has been the focus of many publications during the last 2-3 years, all of them in top-notch journals (Nature, Science). This highlights the interest of the general scientific community for the role of mitochondrial protein import perturbations on cellular (dys)function. However, there are significant differences in these published papers to our work.

- a) *In these studies the means of mitochondrial protein import inhibition is different.* For example, in mitoCPR (PMID: 29650645) the authors choose to overload mitochondria with a precursor with a bipartite mitochondrial targeted signal which specifically impairs import of bipartite signal containing precursors that target the inner mitochondrial membrane. In UPRam (PMID: 26245374) the authors chose to inhibit the MIA pathway of import of intermembrane space proteins, or to overexpress mitochondrial precursors without a targeting signal so as to cause their accumulation in the cytoplasm and monitored the cytoplasmic effects. mPOS (PMID: 26192197) is induced by mutation in the adenine nucleotide translocase (ATP/ADP carrier) which triggers inner membrane dysfunction and leads to the mislocalization of specific precursors and their accumulation in the cytoplasm. MitoTAD (PMID: 31118508) represents a possible mechanism through which precursors are removed from clogged translocases (TOM) under steady state conditions so that the system remains functional. MAGIC (PMID: 28241148) is mitochondrial protein import-facilitated dissolution and degradation of cytosolic aggregated proteins. The existence of such a mechanism suggests that import defects would render cytoplasm prone to aggregation independently of the accumulation of mitochondrial precursors. *None of the above interventions leads to reduced mitochondrial abundance as TOMM-40 and TIMM-23 depletion does.*
- b) Overall, these studies highlight the proteotoxic stress that is imposed in both the cytoplasm and mitochondria upon mitochondrial protein import inhibition. This response in some cases improves the function of stressed mitochondria (like in the case of mitoCPR) while in others it represents a novel mitochondrial cell death pathway (like in the case of mPOS). So *the outcome of each intervention is different, underling the large difference between approaches despite the seemingly similar treatment (mitochondrial protein import perturbation).*
- c) Finally, *these studies are performed in yeast*, a model organism which has been the absolute tool for biochemical mitochondrial protein import studies, but also comes with the limitations of a unicellular organism.

In conclusion, our approach differs significantly for what is already published, it reveals a novel metabolic adaptation that takes place upon MitoMISS, yet, it does not contradict the published results.

The conclusion that MitoMISS extends the lifespan of *C. elegans* independent of the canonical UPR^{mt} is unfortunately not convincing and somehow misleading. In my view, the MitoMISS is more likely to belong to canonical UPR^{mt}, as evidenced with the induction of classical UPR^{mt} reporter genes *hsp-6* and *hsp-60* as well as the ATFS-1 dependency. For the results that “MitoMISS further extends lifespan during mild mitochondrial” dysfunction (Figure 2G and 2H), the use of only 10% of *cco-1* or *atp-3* RNAi (as indicated in the methods) make me doubt that the “mild mitochondrial dysfunction-induced lifespan extension” is already “saturated”. RNAi of *timmm-23* could simply further exacerbate the mitochondrial dysfunction, leading to a more robust UPR^{mt} response and thereby further extend the lifespan (the author should use qRT-PCR and/or western blots to compare the expression of UPR^{mt} genes to clarify this point).

We thank the Referee for this comment as it prompted us to further investigate the differences between the conventional and the MitoMISS-associated UPR^{mt}. The experiments we pursued pertinent to the MitoMISS triggered unconventional UPR^{mt} are described in detail below.

Per the suggestion of the Referee, we investigated whether UPR^{mt} induction in the conditions used for lifespan (Figure 2g and 2h) is not saturated and have an additive effect when *cco-1*, or *atp-3* is combined with MitoMISS. We show that 10% depletion of *cco-1* and *atp-3* does not cause a saturated UPR^{mt} response (as the Referee suspected). However, 90% of *timmm-23* depletion triggered robust *hsp-6* expression. More importantly, combination of both interventions does not further increase UPR^{mt} and thus the additive levels of UPR^{mt} response is not the reason behind the increased longevity of *cco-1;timmm-23* or *atp-3;timmm-23* RNAi (Supplementary figure 4b).

To further support the non-canonical UPR^{mt} induction upon MitoMISS, we have now added more data corroborating the fact that MitoMISS has a different transcriptional output compared to conventional UPR^{mt} (triggered by *spg-7* inhibition). We have added a new main figure (Figure 4) in which we monitor the expression levels of known ATFS-1-dependent transcriptional targets of conventional UPR^{mt}, upon MitoMISS. Notably, we show that although there is a group of genes that behave similarly upon *spg-7* and *tomm-40* depletion, there is another group of genes in which MitoMISS induces opposite responses to conventional UPR^{mt}.

It is known that the developmental timing is important for longevity upon mild mitochondrial stress in nematodes (PMID: 15280428, 21215371). Specifically, mild mitochondrial stress after L4 and throughout lifespan fails to elicit a longevity phenotype (PMID: 21215371). Interestingly, MitoMISS starting from L4, unlike conventional UPR^{mt} stressors, fails to mount an UPR^{mt} response (Supplementary figure 4a), yet, it partially preserves its lifespan promoting effects (Supplementary figure 1h-k).

All the aforementioned findings suggest that UPR^{mt} responses may vary according to the stimulus. Moreover, these findings suggest that MitoMISS induces a distinct and thus “non-conventional” UPR^{mt} response.

For the data that “targeted MitoMISS in the neurons failed to extend lifespan” (Figure 6E), the methods used by the authors is obviously different from the previously paper to knockdown *cco-1* specifically in the neuron (where a transgenic worm line carrying an inverted repeat hairpin (HP) targeting *cco-1* was created) (PMID: 21215371).

To address this interesting point raised by the Referee, and test whether neuronal *cco-1* RNAi can promote lifespan extension in the transgenic *sid-1* mutant strain expressing neuronal SID-1, we performed *cco-1* RNAi 10% or 100% in this genetic background. As shown in figure 7f, neuronal *cco-1* depletion extends lifespan of nematodes, contrary to neuronal MitoMISS. Thus, the method used for neuronal specific RNAi is not the reason why neuron-specific MitoMISS fails to induce longevity.

The initial rationale of the paper it is not clear. The authors decided to impact mitochondrial mass by impairing the mitochondrial protein import machinery. However, they do not give any rationale or reference to justify this approach. One could consider that a more straightforward approach to target mitochondrial mass would be to impact mitochondrial biogenesis. This could be achieved for example by genetically inhibiting TFAM homologue *hmg-5*, or DNA polymerase gamma homologue *polg-1*. Such data are required to make the observations of MitoMISS generalizable.

In our revised manuscript we have clarified the initial rationale and main aim of our study in the introduction section. Mitochondrial protein import represents a major step in the processes of mitochondrial biogenesis (PMID: 30626975). As described in the manuscript we avoided knocking down of transcription factors implicated in mitochondrial biogenesis as this could have broader effects in cellular and organismal physiology. Moreover, *polg-1* mutation leads to critically shortened lifespan in both worms and mice (PMID: 19181702, PMID: 15164064) an effect that is likely due to accumulated mtDNA deletions and mutations and to great imbalance between mtDNA and ncDNA derived precursors. However, we agree with the Referee that we needed to generalise MitoMISS effects on lifespan. To achieve this, we used a known pharmacologic inhibitor of mitochondrial protein import, dequalinium chloride or DECA (PMID: 2537136). Wt nematodes grown in the presence of increasing concentrations of DECA present a dose dependent lifespan extension effect (Figure 1i) corroborating the effects of MitoMISS on longevity.

The use of muscle and intestine-specific mitochondrial reporters to indicate the “mitochondrial mass” may not reflect the reality; the authors should also determine the expression of mitochondrial proteins (e.g. ATP5A, NDUFS3) with western blots during MitoMISS. Also, does MitoMISS affect the mtDNA/nucDNA ratio?

As the Referee suggested we have now included additional data regarding the effects of MitoMISS on mitochondrial mass. Specifically, we now use a single copy translational reporter of COX-4::GFP that was generated by eGFP tagging of the endogenous *cox-4* locus and is expressed ubiquitously (PMID: 3002487). We showed that MitoMISS conditions exhibit the most severe reduction

in COX-4 protein levels (Figure 1c). Moreover, we analyzed the protein levels of the mitochondrial protein ATP-5 in total worm lysates and observed a significant reduction in MitoMISS samples (Figure 1g). Finally, we analyzed the mtDNA/ncDNA ratio by qPCR and showed that interestingly, mtDNA is downregulated in all mitochondrial protein import perturbations (Supplementary figure 1g). Therefore, all these new findings together with the previous ones, obtained from tissue-specific mitochondrial targeted GFP reporters and staining with three different mitochondrial specific dyes, all point to the fact that mitochondrial abundance is significantly reduced upon MitoMISS.

The authors overall relied a lot on the GFP expression intensity of individual reporter strains, they should also use qRT-PCR to verify at least some of the key results (e.g. Figures 2D, 3D, 4A, 4D).

Per the suggestion of the Referee, we have added RT-PCR data to verify key results obtained from transcriptional reporters like *phsp-6GFP* (Supplementary figure 4c), *ppgi-1GFP* (Supplementary figure 5b), *pfgt-1GFP* (Supplementary Figure 5c).

Why do *tomm-40*, *timm-23*, *timm-22* and *gop-3* RNAi's give so different results regarding *ubl-5* induction.

In the revised manuscript and prompted by the suggestion of the Referees we have added a positive control both in the *DVE-1::GFP* and *UBL-5::GFP* experiments. As we didn't manage to activate *UBL-5::GFP* with known UPR^{mt} stressors (*spg-7* RNAi), we removed this panel from the study.

Why *DVE-1* and *UBL-5* are not required for the observed phenotypes is unclear.

DVE-1 is not activated upon mitochondrial protein import inhibition as shown in the revised supplementary figure 4k, and although *dve-1* RNAi treated animals are sick and short-lived, nevertheless, simultaneous knock-down of *dve-1* and *tomm-40* leads to a statistically significant lifespan extension (figure 3f and supplementary figure 4g), something that we never observed with *atfs-1* RNAi animals or *atfs-1* mutants (*ATFS-1* is absolutely required for MitoMISS associated longevity). This finding is what initially triggered us to claim that this is an unconventional UPR^{mt} response, a notion that is supported by many additional data in the revised manuscript described in detail above and in new figure 4, Supplementary figure 4a,b and supplementary figure 1h-k. It is tempting to speculate that the involvement of *DVE-1* could differentiate between different UPR^{mt} transcriptional outputs however more data would be needed for this claim, and this is currently out of the scope of this manuscript.

In line 220-224. The authors make a bold statement by excluding the participation of chaperones and proteases to the observed effect just performing experiments using *gpi-1* RNAi. They should use RNAis against important targets of ATFS-1 to demonstrate this point.

We thank the reviewer for this comment as it prompts us to assay the effect of the classical UPR^{mt} chaperone HSP-60 in MitoMISS longevity. Interestingly, we find that although depletion of HSP-60 caused pronounced shortening of nematode lifespan, MitoMISS can completely rescue this effect (Figure 6g). This finding further strengthens our hypothesis that mitochondrial chaperones are not the main reason for MitoMISS associated longevity. Moreover, it became apparent that MitoMISS can rescue the effects of mitochondria-associated damage.

Minor comments

The figures panels are often referred in the text not following alphabetical (logical) order. This is confusing for the reader and makes it difficult to follow the flow of the story. The authors should properly reference the figure panels in which the data are presented each time. Moreover, the lack of figures numbers in the figure panels exacerbates this issue.

We apologize for these discrepancies. We have put more effort to avoid such mistakes in the revised manuscript. We have also added figure numbers where they were missing.

The authors present the same data in Figure 2F and Figure S1G (in the latter they also include other conditions). It is not standard to show the same lifespan curves in two different figures.

We thank the Referee for pointing this out; we have now corrected it by showing in the revised figure 2f all the conditions (MitoMISS both in *wt* and *aak-2* mutants) in the same panel and by removing the old Figure S1G.

In Figure 2G it would be interesting to know whether *cco-1*(RNAi);*tim23*(RNAi) is statistically different from *cco-1*(RNAi) alone. Same goes for Figure 2H, *tim23*(RNAi);*atp-3*(RNAi) and *atp-3*(RNAi)

We have added the asterisks denoting statistical difference in the revised figure 2g-h. Statistical differences for lifespans are also reported in Supplementary Table 2.

Referee 2:

The manuscript by Lionaki et al investigates the role of mitochondrial mass in *C. elegans* longevity. It addresses important questions that remain unresolved in the context of mitochondrial longevity and the role of the UPR-mt in lifespan extension. The paper shows that RNAi-mediated depletion of the ribosomal import pores tomm-40 and timm-23 depletes mitochondrial content, ROS, and ATP. Knockdown of these genes, but not of other mitochondrial import factors such as timm-22, extends lifespan. This appears to be independent of insulin signalling or lifespan extension mediated by SKN-1. Inhibition of mitochondrial import triggers the atfs-1 dependent induction of hsp-60 and hsp-6 and the longevity is, likewise, atfs-1 dependent. Glycolysis genes were affected by tomm-40 and timm-23 RNAi and this was mirrored by enhanced glucose uptake. Metabolome analysis then confirmed changes in glycolysis and revealed increased steady state levels of serine. Of note, serine supplementation did not further extend the survival of tomm-40 RNAi treated worms. Finally, the authors use tissue specific RNAi and find that knockdown in the body wall muscle and intestine are sufficient for lifespan extension.

The manuscript is concisely and clearly written and documents a number of very interesting observations that stand to enhance the knowledge in the field of mitochondrial longevity and metabolism. However, some of the conclusions are not solidly supported by the data and their interpretation is at times unclear.

We thank the Referee for his/her positive evaluation of our study and his/her very helpful and constructive comments which have led us to experimentally solidify our initial claims and greatly improve our manuscript. The Referee's feedback has been invaluable towards this direction.

A general characterization of the MitoMISS animals would add important insights. Given the low ATP levels, what is the effect on brood size, developmental time and all-over morphology during the knockdown?

As the reviewer suggested we have included in the revised manuscript the effects of mitochondrial protein import perturbation on body size, brood size and developmental timing. Specifically, we show that MitoMISS animals have a lower brood size and body size compared to control (supplementary figure 1d,e). Also they exhibit a small developmental delay of about 6-8 hours, as discussed in the revised text.

It would be important to solidify the MitoMISS phenotype regarding mitochondria. Is there a reduction of mito DNA copy number? Are there changes in biochemical markers of mitochondrial abundance?

We thank the Referee for this comment. As requested, we have now added additional evidence of the reduced mitochondrial abundance in MitoMISS animals. Specifically, we now include

an additional single copy translational reporter of COX-4::GFP that was generated by eGFP tagging of the endogenous *cox-4* locus and is expressed ubiquitously (PMID: 3002487). We showed that MitoMISS conditions exhibit the most severe reduction in COX-4 protein levels (Figure 1c). Moreover, we analyzed the protein levels of the mitochondrial protein ATP-5 in total worm lysates and observed a significant reduction in MitoMISS samples (Figure 1g). Finally, we analyzed the mtDNA/ncDNA ratio by qPCR and showed that interestingly, mtDNA is downregulated in all mitochondrial protein import perturbations (Supplementary figure 1g). Therefore, all these new findings together with the previous ones with the tissue-specific mitochondrial targeted GFP reporters and the stainings with three different mitochondrial specific dyes, all point to the fact that mitochondrial abundance is significantly reduced upon MitoMISS. Last, from the metabolomic analysis, we quantified the concentration of lactate in *tomm-40* (RNAi) treated worms vs. the WT. The observed substantial increase in the lactate abundance upon MitoMISS is compatible with the decreased mitochondrial abundance, in the context of the rest of the measurements (Figure 6b). In the revised manuscript, we have included text explaining this finding and its connection with MitoMISS at the end of the section “De novo serine biosynthesis mediates MitoMISS associated longevity” where we are collectively presenting the findings of metabolomic analysis.

Given that the MitoMISS phenotype is based on RNAi, it would be important to quantify the efficiency of the knockdown of the 4 genes in Figure 1 using qPCR.

The verification of RNAi-mediated mRNA reduction of *timm-23*, *tomm-40*, *timm-22*, *gop-3*, *atfs-1* and *gpi-1* by qPCR has been incorporated in Supplementary figure 1c, Supplementary figure 4d and Supplementary figure 5a. Moreover, the efficiency of *phgdh-1* RNAi is tested in the new translational reporter strain expressing full length PHGDH-1 fused with GFP under its endogenous promoter, shown in Figure 6i.

Is there an induction of GCN-2 in the MitoMISS animals (See Baker et al., 2012)? A GCN-2/eIF2alpha induction might be expected in animals with reduced mitochondria and *atfs-1* induction.

We have tested for eIF2alpha total protein levels and phosphorylation levels upon mitochondrial protein import impairment. We found that although total protein levels of eIF2alpha are significantly increased upon both the lifespan prolonging conditions (*timm-23* and *tomm-40* RNAi) and the ones that don't affect lifespan (*timm-22* and *gop-3*) the ratio of phosphorylated eIF2a/total eIF2a is significantly reduced upon all RNAi treatments as compared to the control condition (Supplementary Figure 3d). Moreover, in new supplementary figure 3, we show that neither the unfolded protein response in the endoplasmic reticulum (UPR^{ER}) nor the heat shock response (HSR) in the cytoplasm are induced upon MitoMISS or any mitochondrial protein import perturbation. To address these questions we used the transcriptional reporters of *hsp-4* and *hsp-16.2* genes (for UPR^{ER} and HSR respectively) and the translational reporter of HSF-1 (for HSR). These data argue against a robust

proteotoxic response outside mitochondria, something that has been suggested before (PMID: 27610574). Since proteostasis pathways outside mitochondrial do not correlate with the observed longevity upon MitoMISS, we chose not to pursue their implication any further.

Generally, in many lifespan assays, the untreated/wild type controls were not included. This is generally acceptable but in Fig 3E, given the absence of an effect, it would be important to establish the effect of the RNAi treatment in the same experiment.

We agree with the Referee on this, and we always include the control conditions in our lifespan experiments. For reasons of clarity of the figure, we added the control conditions with the statistical analysis in Supplementary Table S2.

Fig 4: more experiments are needed to test if really an increased glucose flux is linked to lifespan extension in the MitoMISS animals. A transcriptional reporter for *gpi-1* is not sufficient to conclude that there is increased glycolysis (line 212). The fact that *gpi-1* RNAi alone extends lifespan makes it hard to interpret the data as, in a scenario where *gpi-1* is required for MitoMISS longevity, the survival of the *gpi-1;tomm-40* double treatment should phenocopy the *gpi-1* single-treatment. Here qPCR is needed to check the effect of the double vs single RNAi treatments.

We agree with the Referee that *gpi-1* induction alone cannot prove increased glycolytic flux. Therefore, the expression of additional genes involved in glycolysis was examined. Interestingly, we found that the expression level of glyceraldehyde-3-phosphate dehydrogenase *gpd-2* (glycolytic gene) is increased upon MitoMISS while its expression is not affected upon *timm-22* and *gop-3* RNAi. In this context, we found that expression levels of genes involved in metabolic pathways parallel to glycolysis such as glycogen metabolism and pentose phosphate pathway are also increased upon MitoMISS (*gsy-1* and *gspd-1*) (Supplementary Fig. 5b). To further verify the effect of MitoMISS on glucose levels, we generated transgenic animals expressing Glifon4000, a fluorescent glucose indicator, in pharyngeal muscles (PMID: 30869867). As expected, MitoMISS-treated animals exhibited high levels of glucose (Fig. 5c,d). Moreover, the increase in the glycolytic flux towards *de novo* serine biosynthesis to accommodate a higher rate of fatty acid mobilization has been supported by the metabolomic profiling data (see Figure 6a, b and Suppl. Table 2). More specifically, the increase in the lactate abundance upon MitoMISS is compatible with the decreased mitochondrial abundance. In the revised manuscript, we have included text explaining this finding and its connection with MitoMISS at the end of the section “*De novo* serine biosynthesis mediates MitoMISS associated longevity” where we are collectively presenting the findings of metabolomic analysis.

Regarding the *gpi-1;tomm-40* RNAi longevity, we agree with the Referee that in a scenario where *gpi-1* is required for MitoMISS longevity the lifespan of the double RNAi should be comparable to that of *gpi-1* RNAi alone. We reason that this can be explained by the fact that *gpi-1* RNAi associated longevity is mediated through mitochondrial biogenesis (PMID: 17908557), a criterion that

is not met upon MitoMISS. Furthermore, the efficiency of *gpi-1* and *tomm-40* RNAi from both the single and the double vectors is verified by q-PCR (Supplementary figure 5a).

In a similar argument, *fgt-1* is induced by *tomm-40* RNAi, but is this responsible for the increased glucose uptake (as done in Fig 4C?) In Fig 4D, better fluorescence images are needed, along with DIC images. Can the increase of *fgt-1* expression be confirmed by qPCR?

We have added new images in the respective figure (now figure 5c) trying to better depict the uptake of the glucose analogue dye. Indeed, the brighter part of the tissue is around the intestinal lumen, although the whole tissue exhibits fluorescence. However, exogenous glucose inhibits the uptake of the glucose analogue even in the area of the gut lumen suggesting that the signal is highly specific and represents the incorporated glucose analogue. In an effort to address the same question with a different approach, we constructed a novel glucose reporter strain that expresses the recently published fluorescent glucose indicator, Glifon4000 (PMID: 30869867). Using this reporter we show that endogenous glucose levels are increased upon MitoMISS, corroborating that finding with the glucose analogue dye (Figure 5d). Finally, we confirmed the induction of *fgt-1* by qPCR (Supplementary figure 5c).

Increased lactate and glucose concentrations in Sup Table 2 can be used as an argument for more glycolytic flux. Perhaps it's worth considering changing Figs 4 and 5 to better support the idea of increased glycolysis.

We thank the Referee for this comment. Indeed, the increased glycolytic flux is supported by the metabolic profiling measurements as the Referee suggests. It is actually apparent from the network depiction of Fig. 6b. The original manuscript lacked a clear description of Figure 6b and the metabolomic data analysis results along with the basis for their interpretation. In the revised manuscript, we have included such a short description in the beginning of the section "*De novo* serine biosynthesis mediates MitoMISS associated longevity" appropriately modifying the legend of Figure 6b too. Without changing the sequence of the two sections and Figures as the Referee suggests, we have included a sentence in which we discuss the observed increase in the glycolytic activity based on the metabolic profiling measurements, which supports the gene expression observations discussed in the previous section.

Moreover we have added new gene expression data and transcriptional reporters on other genes associated with glucose metabolism (Figure 4 and Supplementary figure 5) and we show that the main pathways that metabolize glucose are likely activated (pentose phosphate pathway and glycogen synthesis). Finally, we have constructed a new glucose reporter worm strain that express Glifon4000 specifically in the pharyngeal muscles which also points to the fact that glucose concentration is increased upon MitoMISS (Figure 5d). In this way, as the Referee suggests, the two types of measurements corroborate towards the same conclusion and solidify the implied phenomenon. We believe that our efforts adequately address the valid point made by the Referee.

Fig 5C: the authors observe increased serine levels in MitoMISS worms and test if this is required for longevity by suppressing expression of PHGDH. This RNAi treatment by itself extends lifespan. However, *phgdh-1/tomm-40* double knockdown results in WT lifespan. To interpret these data, it would be important to measure serine levels in each of the conditions. Does serine elevation or reduction extend lifespan? Or possibly both? (Does serine supplementation suppress the *phgdh-1* RNAi lifespan extension?)

In the revised manuscript we have added new data corroborating the induction of *de novo* serine biosynthesis. Specifically, we have constructed a translational reporter line which expresses full length *phgdh-1* fused with GFP under its endogenous promoter. We show that MitoMISS induces an increase in PHGDH-1 protein levels in contrast to other mitochondrial protein import perturbations (Figure 6i). This finding strengthens our model that MitoMISS leads to an adaptive metabolic shift encompassing *de novo* serine biosynthesis.

To further investigate the role of exogenous serine supplementation as opposed to *de novo* serine biosynthesis on longevity, we exposed wild type, *tomm-40* depleted and *phgdh-1* depleted animals to increasing concentrations of exogenous serine (5 mM, 25 mM and 50 mM) from hatching throughout adulthood and monitored their lifespan. Interestingly, we noted that in all genetic backgrounds increased concentrations of serine beyond 5mM is detrimental for lifespan (Fig. 6 f-h). TOMM-40 depleted animals are more sensitive to increased concentrations of serine, likely due to the fact that their endogenous serine levels are already increased. Interestingly, *phgdh-1* depleted animals don't show a protection against high concentrations of exogenous serine. The latter suggests that upon inhibition of *de novo* serine biosynthesis animals adapt their endogenous serine levels from dietary serine, and thus exogenous supplementation has the same effects on them as on control animals.

Our findings based on the genetic and metabolomic analysis, have indicated that MitoMISS worms induce the *de novo* serine biosynthesis pathway from glucose. It is the activation of this pathway, indeed, that apparently increases the serine resource for the biological system and enables the worm to keep the activity of the cytoplasmic one-carbon metabolism and other pathways crucial for its survival in light of the decreased mitochondrial abundance and activity. We need to point out that we do believe that it is not the concentration of serine *per se* that indicates the importance of this pathway in lifespan, but the observations in all the metabolite concentrations shown in Fig. 6b that contribute in combination to this finding. In the revised manuscript, we have added a better description of Fig. 6b and how it supports our suggestions for the reprogramming of various metabolic pathways. In this way, our argument may be clearer to the reader.

Fig 6A: the authors state that MitoMISS worms are rescued from glucose toxicity (line 335). I don't fully agree with this interpretation as glucose is still toxic to tomm-40 RNAi treated worms. tomm-40 RNAi still extends lifespan under glucose, but glucose remains toxic. To get a good idea of this effect it would be important to quantify glucose in the various experimental conditions of Fig6A.

In this comment, the Referee asks for concentration measurement of a particular metabolite (here glucose) to allow for conclusions of a more complex pathway and route. In this study, through genetic modifications and the metabolic profiling of worms, we extracted the particular conclusion about glucose toxicity from a combination of metabolite abundance measurements. Glucose is toxic when routed towards pathways that promote inflammation. In this case, we show that upon MitoMISS, despite the high glucose concentration, the worm finds ways to re-arrange its metabolism in light of the decrease in the mitochondrial activity, so that the glucose is appropriately metabolized through routes that do not promote inflammation, which leads to an increase in the lifespan. Regarding the *tomm-40* deficient worms compared to the controls (shown in Figure 6a and 6b and Suppl. Table 2, we have observed an increase in the glucose concentration of the *tomm-40* deficient worms (see dashed red boxed glucose in Figure 6b), but still the *tomm-40* RNAi worms have an increased lifespan. We see that in *tomm-40* deficient worms, glucose is routed through glycolysis towards *de novo* serine biosynthesis, while also contributing to increased myo-inositol concentration/metabolism, thus inositol-based phospholipid metabolism. In addition, we observe increased glucose-trehalose cycle, indicating a “change in sugar storage from glycogen to trehalose”, which has been shown to prevent the worm from the “harmful effects of a high-sugar diet” (PMID: 29511104). Thus, it is not the glucose concentration per se that leads us to this conclusion, but a combination of observations that indicate a vaster metabolic re-programming that leads to increased lifespan. In this regard, we would like to kindly disagree with the Referee that the glucose concentration measurement of the other conditions will change our conclusion based on the current measurement. Nevertheless, the Referee is correct about saying that MitoMISS does not fully rescue from glucose toxicity. Therefore, in the revised manuscript, when mentioned, the word “rescues” has been appropriately replaced by expressions that show amelioration, appease, decreased impact rather than complete exclusion.

The longevity of ETC inhibition shows a unique temporal requirement for lifespan extension within the UPR-mt. At what age is MitoMISS RNAi treatment required for lifespan extension? While I see that this might be beyond the scope it would be a pity not to address such an exciting question in the manuscript.

We agree with the Referee that temporal requirement of MitoMISS intervention is interesting and thus we included such an analysis in the revised manuscript (Supplementary figure 1h-k). It is known that the developmental timing is important for longevity upon mild mitochondrial stress in nematodes (PMID: 15280428, 21215371). Specifically, mild mitochondrial stress after L4 and throughout lifespan fails to elicit a longevity phenotype. Interestingly, MitoMISS starting from L4, unlike conventional UPR^{mt} stressors, fails to mount UPR^{mt} response (supplementary figure 4a), yet, it

partially preserves its lifespan promoting effects (Supplementary figure 1h-k). Although MitoMISS during development maximizes its effects on longevity, it retains part of its pro-longevity effect when imposed during adulthood. These findings strengthen our notion that MitoMISS induces a distinct pathway that mild ETC inhibition, and that induction of mitochondrial chaperones destined to ameliorate mitochondrial proteotoxic stress is not the critical factor for MitoMISS associated longevity.

Minor points:

It would be great to capture better images for Fig. 4C, including DIC images. The signal looks very much like it would be associated with the gut lumen, which is concerning despite the elegant controls with D-Glucose. Please define dex in the figure legend.

We have added new images in the respective figure trying to better depict the uptake of the glucose analogue dye. Indeed, the brighter part of the tissue is around the intestinal lumen, although the whole tissue exhibits fluorescence. However, exogenous glucose inhibits the uptake of the glucose analogue even in the area of the gut lumen suggesting that the signal is highly specific to the incorporated glucose analogue. In an effort to address the same question, in a different approach, we constructed a novel glucose reporter strain that expresses the recently published glucose binding fluorophore, Glifon4000 (PMID: 30869867). Using this reporter we show that endogenous glucose levels are increased upon MitoMISS, corroborating that finding with the glucose analogue dye (Figure 5d). We changed the term dex to d-glucose.

Fig. 5A is not readable and there is generally not enough information for the reader in the figure. Are the changes in metabolite levels significant? What is the meaning of the coloured boxes in Fig 5B?

The purpose of previous Figure 5a (currently 6a) was to show the normalized metabolic profiles provided in Suppl. table 2 in a color-coded way, indicating the distinct metabolic profile between the control and *tomm-40(RNAi)* worms, as clearly supported by the hierarchical clustering analysis. Upon the Referee's recommendation we have now changed Figure 6a to depict the metabolites shown in the reconstructed metabolic network (Figure 6b). In the revised manuscript, Figure 6a encompasses the metabolites in part of the network shown in Figure 6b. As the concentrations of the metabolites differ by many orders of magnitude, the colour-code of the heat map is based on the median of the metabolomic dataset (1.3), with the concentrations below that value appearing as green-coloured and the concentrations above that value as red-coloured. Using this particular colour code, in Figure 6a abundance differences between the two worm conditions are apparent for most of the metabolites, but those in significantly higher than the median abundance in both conditions.

Figure 6b shows the part of the primary metabolic network from glucose to lactate including the serine, glycine and threonine branch. The Referee is correctly pointing out that the legend of the Figure or the text do not describe what the boxes, black or red, are depicting around various

metabolites. In the revised version of the manuscript, we have now modified the legend of Figures 6a and 6b to make them clearer to the reader per the Referee suggestion. Metabolites with significantly higher concentration upon *tomm-40* inhibition are shown in red boxes, dashed red boxes indicate increased abundance in these conditions that cannot be considered statistically significant due to biological variation between replicate samples) while metabolites that remain unchanged are shown in black boxes. In addition, we have added a short description of Figures 6a and 6b in the beginning of the section “De novo serine biosynthesis mediates MitoMISS associated longevity”, that helps the reader understands the results that are discussed in this section about the activity of specific pathways and are based on the data visualized in Figure 6b.

Figures S2C and D are presented without a positive control. It would be important to add one to show the induced state of the assay.

We have added *spg-7* RNAi as a positive control for DVE-1 induction (Supplementary figure 4k). Moreover, we have changed the respective images of the DVE-1::GFP translational reporter to better depict changes in its expression. However, *spg-7* RNAi didn't work properly for UBL-5 thus we removed the relevant data from the study. To further investigate the unconventional UPR^{mt} induction upon MitoMISS, we have now added more data corroborating the fact that MitoMISS has a different expression pattern than conventional UPR^{mt}. We have added a new main figure (Figure 4) in which we monitor the expression levels of known ATFS-1-dependent transcriptional targets of conventional UPR^{mt}, upon MitoMISS. Notably, we show that although there is a group of genes that behave similarly upon *spg-7* and *tomm-40* depletion, there is another group of genes in which MitoMISS induces opposite responses than conventional UPR^{mt}, underlying that different mitochondrial stress signals could elicit slightly different UPR^{mt} responses.

For some of the Figures, the data are organized in a different order compared to the text (e.g. Fig 1C and D). Please mention GST-4 in the results text on page 6. UBL-5 and HAF-1 are not introduced before they are referenced on page 8.

We apologize for these discrepancies. We have put more efforts to avoid such mistakes in the revised manuscript. We have also added figure numbers.

There are two Supplementary Table 2 files

We thank the Referee for noticing. We have corrected this mistake.

In the Supplementary Table with the lifespan data it would be good to show which experiments were done as a group at the same time. Please include in the figure legend the number of repeats for each lifespan.

We used shades to highlight conditions that run simultaneously. We need to point out that the main conditions of MitoMISS in wt and *atfs-1* mutant conditions have been performed many more times than the ones depicted in Supplementary table 2.

Nomenclature of molecules should be the same in text and figures (glycerate-3P vs 3-phosphoglycerate, line 291 and Fig5B).

We want to thank the Referee for the meticulous work on our manuscript which helped us to edit it and improve it.

In closing, we would like to thank the Referees for the constructive and positive input that has enabled us to significantly improve our paper. We do hope that you will find our revisions adequate for publication of our study in *Nature Communications*.

Thank you for your consideration of our manuscript. We look forward to hearing from you in the near future.

Sincerely,

Nektarios Tavernarakis,

For the authors

Reviewers' Comments:

Reviewer #1:

Remarks to the Author:

Despite the additional efforts to support the proposed "novel" MitoMISS concept, the newly added results and the point-to-point responses unfortunately failed to convince me that the proposed "novel" MitoMISS acts differently from the canonical UPR^{mt} or proteotoxic stress signaling. The revised manuscript still relied too much on the GFP expression intensity of individual worm reporter strains and the quality of many of the new provided results are unfortunately too low to support their conclusions.

First, most of the new results added were of low quality. For example, the eif-2a WB data, where the changes of the ratio of phosphorylated eIF2alpha to total eIF2alpha are clearly mostly dependent on the changes of the total eIF2alpha changes but not the P-eif2a level, hardly convincing (Suppl Figure 3). Many new results were heavily relying on the GFP-images of worms, e.g. the new results in all panels of Figure 4, which is not reliable as one can easily notice the huge Standard Deviation within each group. Additionally, the PHGDH-1::GFP results in Figure 6i also suffered from the inconsistency of the individual GFP-worm data as well, e.g. the two worms in the same "control RNAi" condition demonstrated totally different GFP level of PHGDH-1::GFP; so does the results related to the combination of timm-23 and cco-1/atp-3 RNAi data.

Second, the authors have included new metabolomics data and pointed out that "De novo serine biosynthesis mediates MitoMISS associated longevity". However, the quality of the metabolomics measurements is of significant concerns, as the number of samples for each condition (n = 2) and the barely observed difference in each measured metabolite between the control and tomm-40 RNAi worms, do not seem to support any statements regarding the serine synthesis in mediating the MitoMISS.

Finally, these investigators failed to reproduce their own lifespan results shown in different panels of the same figure. It seems that 5 mM serine strongly extended the lifespan of control worms in Figure 6d, but only demonstrated very minor effect in the lifespan curve in Figure 6f. In addition, if serine biosynthesis is required for MitoMISS-induced longevity, then you would expect that supplementation of exogenous serine should extend the lifespan of phgdh-1 RNAi fed worms to the extent as the tomm-40 RNAi fed worms. However, as shown in Figure 6e, serine supplementation did not rescue this phenotype (dashed red line vs. green line), suggesting that serine biosynthesis is not the key for MitoMISS. The authors again failed to convince the differences between the metabolic rewiring happening in MitoMISS and the subsequent/downstream metabolic changes upon the canonical UPR^{mt}.

Therefore, with the current results the revised manuscript contained, I unfortunately could not support its publication in Nature Communications.

Reviewer #2:

Remarks to the Author:

The authors have added significant data and have addressed my concerns. They have added significant data that solidify their original claims. Additional questions remain open, such as how atfs-1 is involved in both the canonical and the non-canonical UPR^{mt}. However, the MitoMISS-dependent non-canonical UPR^{mt} is well supported by the data, for example by looking at the temporal dynamics. I support the publication of the paper.

Reviewer #3:

Remarks to the Author:

In this study, Lionaki et al. argue that suppression of mitochondrial protein import (MitoMiss) leads to longevity through ATFS-1-dependent unfolded protein response. This leads to metabolic reprogramming, increased glucose uptake and upregulation of glycolysis. Later the authors

connect upregulation of glycolysis to increased serine biosynthesis branch from glycerate-3P and claim that de novo serine biosynthesis is required for MitoMiss-induced longevity. Although, the specific mechanism is not explored, the reported metabolomics and RNAi observations seem to support the general conclusion that the branch of glycolysis to de novo serine biosynthesis and the downstream pathways are required for MitoMiss-induced longevity. The metabolomics data is supportive of this general conclusion; however, the analysis and presentation of metabolomics data has several significant issues that need to be addressed.

Major points:

- Out of 73 metabolites listed in the Supplementary Table 3, 65 metabolites are increased and only 8 metabolites are decreased for tomm-40(RNAi) vs. control. It is usually not expected that upon a perturbation, the whole metabolism shift in one direction (up or down), changes in metabolism are usually expected to be balanced, some pathways go up some other pathways go down. It seems that authors need to apply a median normalization on the metabolomics data i.e. normalize signals of all metabolites in each sample to the median signal of that sample. After applying median normalization, 32 metabolites are increased, and 41 metabolites are decreased for tomm-40(RNAi) vs. control which seems more reasonable and balanced. The authors should use the median normalized data for any downstream analysis.
- It is not super clear what metric is used for hierarchical clustering on Fig. 6A. The scale goes from 0 to 13. The figure legend states: "The colour-code shows in green and red colour, respectively, the metabolite abundances that are below or above the median of the metabolomic dataset". What does the "median of metabolomic dataset" indicate? Is it median of all signals measured for all replicates and all metabolites? Regardless, the representation of heatmap could significantly be improved to make interpretations easier and more meaningful. It is preferred that a balanced color-scale be used where red and green colors relatively equally represent magnitude of change in either direction. For example, for values of each row, the authors should subtract row mean from each row value and divide by standard deviation of the row. This will result in positive and negative values for metabolites. Then the authors should pick the highest absolute value from the heatmap (e.g. 10) and set the color-scale from -10 to +10 centered at 0. This will enable a better distinction between metabolic profiles of samples. In addition, sorting the metabolites alphabetically is not the best way to represent a heatmap of metabolites. The authors should either try clustering both rows and columns to let the clustering algorithm decide the proximity of metabolites and samples, or the authors should group metabolites based on their metabolic pathways.
- It is not clear what statistical test was used for metabolomics data (e.g. t-test). The authors should clearly explain the statistical significance test. Methods section states: "Any SAM analysis was performed using as significance threshold (δ , delta) the smallest value that corresponded to a zero false discovery rate (FDR) – median". It is not clear what δ value indicates. A zero FDR is not mathematically possible, FDR is reported as less than 5%, for example. Authors should report the FDR threshold that was chosen to determine significant vs. non-significant metabolites.
- The authors have not measured metabolic flux. Metabolic flux is measured using heavy isotope-labeled nutrients such as ^{13}C -glucose and by performing mathematical flux analysis. Authors should avoid language such as: "Combining the differential metabolite information, we could imply an increased glycolytic flux, as supported by the increased concentration of glycerate-3-phosphate (glycerate-3P) and other downstream pathways". The metabolomics data presented here, shows increased levels of glycolytic metabolites for tomm-40(RNAi) vs. control (although as mentioned above, metabolomics data requires more appropriate normalization), but no information about metabolic flux can be interpreted from this data.

Minor points:

- Extent should change to extend in abstract
- Supplemental table 3 is named supplemental table 2 on the Excel sheet.

July 8, 2021

Ref: *Nature Communications* ms NCOMMS-20-31240A-Z

Dear Referees,

In the following pages, we provide a detailed response to all comments and suggestions.

In the revised manuscript we provide further new insight, relevant to the distinction between MitoMISS-associated, and conventional UPR^{mt}.

Moreover, beyond what has been requested, we have also performed new metabolomic profiling experiments, which address the points raised by both Referees 1 and 3.

A point-by-point response to all the comments follows below (original comments are quoted in **bold**).

Thank you for your consideration of our manuscript.

Sincerely,

Nektarios Tavernarakis,
for the authors

Reviewer #1:

Despite the additional efforts to support the proposed "novel" MitoMISS concept, the newly added results and the point-to-point responses unfortunately failed to convince me that the proposed "novel" MitoMISS acts differently from the canonical UPR^{mt} or proteotoxic stress signaling. The revised manuscript still relied too much on the GFP expression intensity of individual worm reporter strains and the quality of many of the new provided results are unfortunately too low to support their conclusions.

We would like to thank the Reviewer for his/her comments throughout the reviewing process, as they have prompted us to extend our study and better characterize our model. In the following text, we are addressing, point by point, the concerns on MitoMISS-driven cytosolic proteotoxic stress and the MitoMISS-driven UPR^{mt} axis. Moreover, we now include new figures and information, in the revised manuscript that provide further clarifications and support of our findings.

Regarding proteotoxic stress signaling:

In our previous submission, we had checked HSF-1, *hsp-16.2* and *hsp-4* expression levels upon depletion of each of the four mitochondrial import translocases. We had found that neither of the import translocases could induce the expression of the aforementioned reporters. In the current submission, apart from the fluorescent reporter strains of HSF-1, *hsp-16.2* and *hsp-4* (Supplementary Figure 3a-c), we now additionally provide qRT-PCR data of proteasomal, cytosolic, ER and mitochondrial chaperone genes (*pbs-5*, *rpn-6*, *hsp-12.6*, *hsp-70* cytoplasmic, *hsp-3*, *hsp-4*, *hsp-16.2*, *hsp-16.41*, spliced *xbp-1*, *dnj-21*) (Supplementary Figure 3d). Interestingly, we saw that their expression either remained stable or was reduced upon MitoMISS, compared to control, further supporting our initial findings that MitoMISS acts independently of the proteotoxic stress signaling.

Previously, published work from the Morimoto lab, on mild mitochondrial stress and proteostasis in *C. elegans*, unequivocally showed that disruption of several mitochondrial functions, including protein import, primes worms for a much more robust heat stress response (HSR) after heat stress (HS). However, the same mitochondrial perturbations prior to HS, did not induce profound HSR (PMID: 29117555). Moreover, in another worm study from the Dillin lab on mitochondrial-to-cytosolic stress response (MCSR) (PMID: 27610574), it was shown that mitochondrial protein import inhibition does not trigger *hsp-16.2* induction. Therefore, our study is in agreement with previously published studies in *C. elegans*, in terms of how mitochondrial protein import affects cytosolic proteostasis. Taking all the above into account, we concluded that cytosolic proteostatic stress is uncoupled from MitoMISS.

In *Saccharomyces cerevisiae*, acute inhibition of the mitochondrial import in several ways and technical approaches leads to a severe proteotoxic burden in the cytoplasm. This is expected and well documented in previously published works, as noted by the reviewer. However, these perturbations do not lead to reduced mitochondrial load. Specifically, they describe the acute inhibition of the mitochondrial import and assembly pathway (MIA) that mediates import into the mitochondrial intermembrane space has been shown to induce accumulation of misfolded protein into the cytosol

which leads to activation of the so called UPR_m (unfolded protein response activated by mistargeting of proteins). Shifting the temperature sensitive mutant of Mia40 to the non-permissive temperature lowered the abundance of the MIA pathway targets, however, abundance of other mitochondrial proteins was not affected (PMID: 26245374). Moreover, import of matrix targeted precursors is not perturbed in Mia40 mutant mitochondria. Therefore, prompt Mia40p impairment does not lead to a general reduction in mitochondrial load, in sharp contrast with MitoMISS. UPR_m includes a profound increase of the proteasomal genes, proteins, and activity as the proteasome is involved in the degradation of the mistargeted mitochondrial proteins in the cytoplasm. In the same publication, the authors tested the effects of Tim23 or Pam depletion with temperature sensitive yeast mutants. Again, they monitored induction in proteasomal activity after shifting cells to the non-permissive temperature for a few hours. Interestingly, MitoMISS does not induce proteasomal genes as we show in the revised manuscript by qRT-PCR (Supplementary Figure 3d).

How is this system different from ours? In *C. elegans*, we can subject animals to RNAi and suppress the expression of specific, essential genes for long periods of time. This suppression is not complete; it only lowers the expression levels of target genes down to 10-50% of its normal expression levels. This approach allows animals to develop and reach adulthood, even upon impairment of essential genes. During *C. elegans* development, the mitochondrial network experiences a significant expansion during the L3 to L4 stage of development. Suppression of protein import components during this period, abolishes mitochondrial network expansion and thus the total abundance of the functional organelles is lowered. We believe that the difference between MitoMISS worms and mitochondrial protein import mutants of *Saccharomyces cerevisiae* lies on the fact that worms have adjusted to lower mitochondrial mass and probably don't produce as many mitochondrial precursors as animals/cells with fully developed mitochondrial network. The latter suggests that conditions that reduce the mitochondrial protein import rate, without fully compromising mitochondrial import, sustain a smaller but functional mitochondrial fraction with active translocase machineries, ultimately leading to beneficial outcomes for organismal physiology. The tissue of the impairment is also a very significant parameter, as some tissues cannot endure this reduction (neurons) while others, (the metabolic tissues of the animal) actually benefit from such a treatment. Our findings reveal a novel beneficial role of reduced mitochondrial load on organismal physiology, something that was not possible with the excellent published studies in *Saccharomyces cerevisiae*.

Our new findings strongly support that there is no robust cytosolic proteotoxic stress response upon MitoMISS, therefore, proteotoxic stress signaling is quite unlikely to be responsible for the increased lifespan.

Regarding the MitoMISS-driven non-conventional UPR^{mt}:

In our previous submission, the main reason we stated that MitoMISS induces a non-conventional UPR^{mt} is the fact that three key UPR^{mt} players are either not induced or not required for MitoMISS-related lifespan extension (DVE-1, HAF-1, HSP-60) (PMIDs 32934238, 20188671) (Figure 3f,g and Supplementary Figure 4g-j). For instance, genetic inhibition of *tomm-40* was sufficient to extend lifespan of *haf-1* or *dve-1* mutants suggesting that MitoMISS-induced longevity is HAF-1- and

DVE-1-independent (Figure 3f and Supplementary Figure 4g). Moreover, while RNAi suppression of *hsp-60* reduces lifespan of otherwise wild type worms, combined inhibition of *hsp-60* and *tomm-40* completely rescues the phenotype of *hsp-60* deficient animals (Figure 3g). The latter suggests that MitoMISS-associated longevity does not require the conventional players of UPR^{mt} signaling.

Prompted by the comments of the Reviewer, we looked deeper into understanding how MitoMISS is different from canonical UPR^{mt}. Therefore, we focused on known UPR^{mt} targets and tested their expression levels upon MitoMISS and *spg-7* RNAi, a typical UPR^{mt} stressor. We found that silencing of *tomm-40* and *spg-7* engenders a different transcriptional outcome. In the revised manuscript, we combined fluorescent reporter with qRT-PCR analysis, and found that TCA genes (*cts-1*, *mdh-2*, *aco-2*, *icl-1*) emerged as differentially regulated upon *spg-7* and *tomm-40* RNAi, while genes involved in glycolysis (*gpi-1*, *gpd-2*, *gpd-3*, *enol-1*, *aldo-1*) are induced in both conditions. In a recent study, a multi-omic approach with several mitochondrial stressors in mammalian cancer cell lines showed that each stress elicits different responses (PMID: 28566324). Interestingly, they all seem to converge on a few pathways namely, “biosynthesis of amino acids, serine-glycine-threonine pathway” and “carbon metabolism”, among others, through the activation of the integrated stress response (ISR). Our study shows that, although MitoMISS (reduced mitochondrial load) does not induce a typical UPR^{mt} response, it triggers a metabolic shift to glucose metabolism and *de novo* serine biosynthesis, similar to mammalian ISR. Apart from characterizing for the first time the metabolic shift upon UPR^{mt} in invertebrates, our study is the first to uncouple mitochondrial proteostasis from UPR^{mt}-associated longevity, and provide a causative link between the concurrent metabolic shift to the beneficial effects of UPR^{mt} on organismal physiology. These findings are original and extend the role of UPR^{mt} beyond the currently established knowledge.

Notably, recent findings, published while this study was under revision, implicate ATFS-1 in the expansion of the mitochondrial network during *C. elegans* development. Specifically, it was proposed that ATFS-1 is excluded from mitochondria during development as its import is antagonized by the highly expressed mitochondrial precursors, with strong mitochondrial targeting signals. Therefore, ATFS-1 accumulates in the nucleus triggering the expression of several mitochondrial components, participating in mitochondrial network expansion (PMID: 33473112). These intriguing new findings provide a putative underlying explanation for the non-conventional axis of MitoMISS-related UPR^{mt}, as the ATFS-1-dependent expansion of the mitochondrial network is inhibited upon MitoMISS conditions. It is tempting to speculate that reduced mitochondrial protein import during development could actually lead to suppression of mitochondrial genes, thereby reducing mitochondrial biogenesis in total. This intriguing idea would explain the absence of a generalized cytosolic proteotoxic stress. We intend to look into this idea by performing RNA-seq analysis of control and MitoMISS animals, in a future work.

First, most of the new results added were of low quality. For example, the eif-2a WB data, where the changes of the ratio of phosphorylated eIF2alpha to total eIF2alpha are clearly mostly dependent on the changes of the total eIF2alpha changes but not the P-eif2a level, hardly convincing (Suppl Figure 3).

As described in the manuscript and the rebuttal letter, our eIF2 α western blot shows that there is clearly no induction of eIF2 α phosphorylation, per se, as the induction of the phosphorylated form follows an induction of the total eIF2 α . Rather we report a trend towards a reduction in the ratio. Therefore, we concluded that there is not a robust proteostatic response. Per the suggestions of the Reviewer, and in order to avoid any misunderstandings, we have included a graph with the quantified results of three independent experiments in the revised version of our manuscript (Supplementary Figure 3e). In conclusion, we are not in disagreement with the reviewer regarding eIF2 α phosphorylation.

Many new results were heavily relying on the GFP-images of worms, e.g. the new results in all panels of Figure 4, which is not reliable as one can easily notice the huge Standard Deviation within each group.

Regarding this comment of the Reviewer, we would like to underline that the panels in figure 4 are not scatter plots indicating the mean and standard deviation of each value as the reviewer suggests. In the previous revision of our manuscript we opted to present some of the old and many of the new data in “box and whiskers” plots. These plots depict the median, the maximum and the minimum values, excluding outliers. Thus, the lines extending from the boxes indicate the difference between the maximum and minimum values and not the standard deviation, as suggested by the Reviewer. The upper and lower lines of the boxes indicate the median of the upper half of the dataset (third quartile) and of the lower half of the dataset (first quartile), respectively. It is the boxes – not the whiskers – that indicate the main variability in the respective value, related to the standard deviation discussed by the reviewer.

However, to further improve depiction of fluorescence reporter data, we now present the quantified results with violin plots. Unlike bar graphs with means and error bars, violin plots contain all data points. Violin plots are perfectly appropriate even if the data do not conform to normal distribution. They work well to visualize both quantitative and qualitative data. Considering this difference in the information provided in the plots compared to the scatter-plots, we believe that the reviewer may reconsider the quality of our data and figures. We have now changed figure legends to make this difference in the plots clearer.

Regarding the fact that a significant part of our results and conclusions derive from GFP-strains analysis, this may not be considered negative in worm studies. This is actually one of the advantages of working with this model. It gives us the opportunity to follow tissue specific gene expression in a spatiotemporal manner using fluorescent reporters, providing clear information that is not directly possible with other systems. This type of imaging analysis has been routinely used for

several seminal studies in the field of UPR (PMID 27610574, 31974253, 31412237, 24662282). Moreover, in the revised manuscript, our claims are further supported by qRT-PCR analysis (Figure 4j, Supplementary Figure 3d, Supplement Figure 4c and d, Supplementary Figure 5a,b and d) as the reviewer suggested.

Additionally, the PHGDH-1::GFP results in Figure 6i also suffered from the inconsistency of the individual GFP-worm data as well, e.g. the two worms in the same “control RNAi” condition demonstrated totally different GFP level of PHGDH-1::GFP; so does the results related to the combination of *tim-23* and *cco-1/atp-3* RNAi data.

It is expected, for different animals in a population, to display different gene expression or protein levels. We wouldn't want to beautify our findings by choosing animals that express exactly similar gene or protein expression levels, to the mean value of our population. When we monitor gene/protein expression levels with reporter strains/lines we can follow this variability more precisely. The violin plots that accompany the representative images in the revised manuscript, describe more accurately the distribution of the fluorescence values of each population. This variability is depicted also in the representative images of our experiments.

Besides the expected variability, differently treated populations in both figures discussed by the Referee show statistically different protein or gene expression levels as established by the one-way ANOVA with Tukey's multiple comparisons test.

Second, the authors have included new metabolomics data and pointed out that “De novo serine biosynthesis mediates MitoMISS associated longevity”. However, the quality of the metabolomics measurements is of significant concerns, as the number of samples for each condition (n = 2) and the barely observed difference in each measured metabolite between the control and *tomm-40* RNAi worms, do not seem to support any statements regarding the serine synthesis in mediating the MitoMISS.

We would like to kindly disagree with the Reviewer. His/her statement is based on a misunderstanding. No new metabolomic data were included in the previously submitted manuscript. This was the same metabolomic dataset that we provided in the original manuscript, but per the suggestion of Reviewer 2, we changed the presentation of the data in the heatmap form. We have now repeated the metabolomic analysis of another set of worm populations (we have in total 4 repetitions per condition), and all results are validated. This is very important, as the studies are independent, the collection of the samples and the metabolomic analysis took place at two different time periods independently and still the results are the same validating our original findings. The new metabolomic data are provided in the new version of the manuscript in Suppl. Table 3, while the metabolites with statistically different representation between the two populations are summarized in Suppl. Table 4 of the revised manuscript. We would like to re-iterate that the activity of the *de novo* serine biosynthesis pathway has been supported from specific mutants that disrupt specific pathways

and the acquired measurements are the expected only if the *de novo* serine biosynthesis pathway is active. This is indeed a novel result and is of value to make this connection with MitoMISS in the worm model.

Finally, these investigators failed to reproduce their own lifespan results shown in different panels of the same figure. It seems that 5 mM serine strongly extended the lifespan of control worms in Figure 6d, but only demonstrated very minor effect in the lifespan curve in Figure 6f.

The observed variation in the life span between different experiments and populations is well expected and within the expected range for this type of experiments. This variation does not contradict our findings and does not diminish their value. In the revised manuscript we include a forth repetition of experiment (depicted in new figure 7d). Below, the four repetitions of the aforementioned conditions, control and 5mM of serine are depicted (all of them are included in Supplementary table 2, summarizing the lifespan data).

Figure 1 of response letter: Lifespan curves of all four repetitions of control population vs worms that have been reared in the presence of 5mM L-serine supplementation. In all repetitions L-serine supplementation significantly extends lifespan. Statistical analysis was performed with the Log-rank (Mantel-Cox) test. a and c, are presented in figure 7a (old figure 6d) and the new figure 7d (didn't exist in the previous submission). b and d, are both only in the new suppl. Table 2 (b is the old figure 6f). The statistics of all repetitions are summarized in table S2.

As noticed, in all repetitions exogenous addition of 5mM L-serine extends lifespan of wt worms, in a statistically significant manner. Moreover, the effects of exogenous L-serine on control population has been previously published (PMID 25643626). We thank the Reviewer for acknowledging every lifespan experiment is different and can be severely affected by the microenvironment of the aging population (humidity, slight temperature fluctuations, NGM plate batch etc.). It is not uncommon to notice such differences between different lifespan experiments therefore we always include control conditions in every experiment that we perform. Indeed, such differences have been reported many times in literature. As an example, we present part of the lifespan data tables of PMID 28853436 (Figure 2 of response letter) and of PMID 24630720 (Figure 3 of response letter). Such differences in median lifespan between repetitions are very common and expected, as far as the trends (extension vs shortening) do not change.

Supplementary Table 1: Lifespan Analyses

Strain/Treatment	Median Lifespan	Median Difference from Control (%)	Worms ⁺	P value	Reference Control.
N2	26		91/120		
eat-2(ad465)	28	+7.69	68/120	<0.0001	vs N2
ncl-1(e1942)	24	-7.69	75/120	ns	vs N2
ncl-1(e1865)	24	-7.69	98/120	0.03	vs N2
eat-2;ncl-1(e1942)	21	-25	40/120	<0.0001	vs eat-2
eat-2;ncl-1(e1865)	16	-42.85	72/120	<0.0001	vs eat-2
N2	25		100/120		
eat-2(ad465)	30	+20	48/120	0.002	vs N2
ncl-1(e1942)	25	0	83/120	ns	vs N2
ncl-1(e1865)	25	0	91/120	ns	vs N2
eat-2;ncl-1(e1942)	25	-16.66	43/120	<0.0001	vs eat-2
eat-2;ncl-1(e1865)	17	-43.33	79/120	<0.0001	vs eat-2

Figure 2: Lifespan data from PMID 28853436. As noticed, differences from the control can differ from +7.69% to +20% for *eat-2(ad465)* mutant, or from -7.69% to 0 for *ncl-1(e1942)* or *ncl-1(e1865)* mutants.

Table S1. Lifespan Data, Related to Figs. 2, 4, 5, 6, S2, S4

Strain	Treatment (RNAi)	Median LS	Difference from cntl	Animals	Max LS	Difference from cntl	P value	Reference cntl
N2 (WT)		21		104/150	32			
gfat-1(dh784)		24	14.29%	108/150	38	18.75%	<0.0001	vs. N2
gfat-1(dh785)	→	24	14.29%	100/150	40	25.00%	<0.0001	vs. N2
N2		21		88/150	31			
gfat-1(dh784)		25	19.05%	63/150	41	32.26%	<0.0001	vs. N2
gfat-1(dh785)	→	23	9.52%	88/150	39	25.81%	0.0004	vs. N2
N2		19		79/150	30			
daf-2(e1368)		30	57.89%	102/150	53	76.67%	<0.0001	vs. N2
gfat-1(dh468)		24	26.32%	94/150	38	26.67%	<0.0001	vs. N2
gfat-1(dh784)		24	26.32%	88/150	42	40.00%	<0.0001	vs. N2
gfat-1(dh785)	→	21	10.53%	85/150	40	33.33%	0.005	vs. N2

Figure 3: Lifespan data from PMID 24630720. As noticed, the median of the control population may differ between experiments whilst differences from control i.e. for *gfat-1(dh784)* mutant, can range from 14 to 26%.

In addition, if serine biosynthesis is required for MitoMISS-induced longevity, then you would expect that supplementation of exogenous serine should extend the lifespan of *phgdh-1* RNAi fed worms to the extent as the *tomm-40* RNAi fed worms. However, as shown in Figure 6e, serine supplementation did not rescue this phenotype (dashed red line vs. green line), suggesting that serine biosynthesis is not the key for MitoMISS. The authors again failed to convince the differences between the metabolic rewiring happening in MitoMISS and the subsequent/downstream metabolic changes upon the canonical UPRmt.

We would like to kindly state that this statement is based on a misunderstanding of the provided information in our manuscript. The condition that the Reviewer describes (*phgdh-1* RNAi) was not depicted in this figure (old Figure 6e, new Figure 7c). What we show in another figure (old figure 6h, new Figure 7f) is that *phgdh-1* RNAi-fed worms actually benefit from exogenous serine, as the Reviewer predicts. Therefore, we believe that there is a confusion regarding this point. In case further clarifications are needed, or our experimental design and results are not clearly depicted, as implied by the Reviewer's comment, we would gladly provide more information.

Reviewer #3):

In this study, Lionaki et al. argue that suppression of mitochondrial protein import (MitoMiss) leads to longevity through ATFS-1-dependent unfolded protein response. This leads to metabolic reprogramming, increased glucose uptake and upregulation of glycolysis. Later the authors connect upregulation of glycolysis to increased serine biosynthesis branch from glycerate-3P and claim that *de novo* serine biosynthesis is required for MitoMiss-induced longevity. Although, the specific mechanism is not explored, the reported metabolomics and

RNAi observations seem to support the general conclusion that the branch of glycolysis to *de novo* serine biosynthesis and the downstream pathways are required for MitoMiss-induced longevity. The metabolomics data is supportive of this general conclusion; however, the analysis and presentation of metabolomics data has several significant issues that need to be addressed.

We would like to thank the reviewer of his/her comments and appreciation of our results supporting the association of the *de novo* serine biosynthesis with the MitoMISS-induced longevity. His/her comments about the presentation of the metabolomics results are addressed below.

Major points:

Out of 73 metabolites listed in the Supplementary Table 3, 65 metabolites are increased and only 8 metabolites are decreased for tomm-40(RNAi) vs. control. It is usually not expected that upon a perturbation, the whole metabolism shift in one direction (up or down), changes in metabolism are usually expected to be balanced, some pathways go up some other pathways go down. It seems that authors need to apply a median normalization on the metabolomics data i.e. normalize signals of all metabolites in each sample to the median signal of that sample. After applying median normalization, 32 metabolites are increased, and 41 metabolites are decreased for tomm-40(RNAi) vs. control which seems more reasonable and balanced. The authors should use the median normalized data for any downstream analysis.

We thank the reviewer for his/her comment. Indeed, as he/she mentions the metabolic physiology may be “symmetrical”, in the case of metabolic reaction fluxes though – the direct metabolic equivalent of transcript and protein expression - not necessarily metabolite abundances *per se*, but this is true for the entire metabolism and for a well-controlled/closed system with the same rate of substrate consumption between the various physiological conditions.

In our case, it is only the primary metabolism that is captured through the metabolic profiling and early secondary metabolism for some hydrocarbons and fatty acids. In addition, in metabolomic analysis, the abundance of each metabolite is quantified per gr of the biological system (here the worm pellet) and not per gr of metabolite extract, to measure the change in the composition of the particular extract in the various metabolites. In an “open” system, with varying the amount of substrate(s) consumed from the “outside” environment (in the case of worms the substrates are indirectly consumed from the bacteria consuming glucose), the fraction of the primary and secondary metabolites (lipids, hydrocarbons) produced and accumulated in the biomass of the particular biological system cannot be considered constant between the various physiological conditions (as we could assume for DNA, RNA). Thus, an increase or decrease in the abundance of most primary metabolites quantified is to be expected depending on the change in the consumption rate of substrates (see relevant statement on pages 270-271 in PMID: 18958862)

This is the case in our analysis too, where the two worm population samples (control and *tomm-40* RNAi) had been made of same weight. However, all accompanying results were indicating

an increase in the glucose consumption rate in *tomm-40* RNAi compared to the control worms, and we observed a significant increase in the total weight of the 73 free metabolites quantified by GC-MS metabolomics and included in the analysis in the *tomm-40* RNAi compared to the control worm population. In the context of this overall increase in the 73 free metabolites, SAM method identified only positively significant metabolites. There are metabolites as the reviewer suggests, the relative abundance of which in the total of the 73 measured metabolites is lower in the *tomm-40* RNAi worms compared to the controls, but very few show a decrease in their actual abundance in the *tomm-40* RNAi worms compared to the controls (if out of the context of the glycolysis and *de novo* serine biosynthesis pathway no further discussion is made in this manuscript).

In the revised manuscript, we have repeated the experiment independently (collection time 2, CT2) and we have obtained similar results.

However, we fully accept the Reviewer's comment and concern, and in the revised manuscript, we provide further information to the reader about the SAM method and the identification of significant metabolites in both collection times (CT1 and CT2). More specifically, in Suppl. Table 3 for both repetitions of the experiment, we have now included apart from the normalized relative peak areas (RPAs) per 100mg of worm pellet (as provided now) in Part A, also the fraction of each metabolite in the total abundance of the metabolites included in each analysis (Part B). Thus, differences in the relative composition of these metabolites in the total quantified metabolite abundance can be apparent. In addition, in Supplementary Figures 6c and 7c we now provide, respectively, the SAM method curves for the two independent repetitions and have added the names of the metabolites on the left part of the curve, which show decrease in their concentration in the *tomm-40* RNAi with respect to the control, but which are not identified as statistically significant by SAM method in the context also of the overall increase in the total quantified free metabolite abundance (positive intercept with y axis).

It is not super clear what metric is used for hierarchical clustering on Fig. 6A. The scale goes from 0 to 13. The figure legend states: "The colour-code shows in green and red colour, respectively, the metabolite abundances that are below or above the median of the metabolomic dataset". What does the "median of metabolomic dataset" indicate? Is it median of all signals measured for all replicates and all metabolites? Regardless, the representation of heatmap could significantly be improved to make interpretations easier and more meaningful. It is preferred that a balanced color-scale be used where red and green colors relatively equally represent magnitude of change in either direction. For example, for values of each row, the authors should subtract row mean from each row value and divide by standard deviation of the row. This will result in positive and negative values for metabolites. Then the authors should pick the highest absolute value from the heatmap (e.g. 10) and set the color-scale from -10 to +10 centered at 0. This will enable a better distinction between metabolic profiles of samples. In addition, sorting the metabolites alphabetically is not the best way to represent a heatmap of metabolites. The authors should either try clustering both rows and columns to let the

clustering algorithm decide the proximity of metabolites and samples, or the authors should group metabolites based on their metabolic pathways.

Per the Reviewer's suggestion, we have now changed Figure 6a (and b) to include the hierarchical tree of the positively significant metabolites for each independent experiment (collection time) based on Euclidean distance, using color-coding on their standardized relative peak areas (RPAs), as the reviewer suggests. The hierarchical clustering of the metabolite RPAs and standardized RPAs based on Euclidean distance, and the SAM curves for both collection times are shown in Supplementary Figure 6a-c, and Supplementary Figure 7a-c, respectively. The full list of positively significant metabolites for both collection times are shown in Supplementary Table 4.

It is not clear what statistical test was used for metabolomics data (e.g. t-test). The authors should clearly explain the statistical significance test. Methods section states: "Any SAM analysis was performed using as significance threshold (δ , delta) the smallest value that corresponded to a zero false discovery rate (FDR) – median". It is not clear what δ value indicates. A zero FDR is not mathematically possible, FDR is reported as less than 5%, for example. Authors should report the FDR threshold that was chosen to determine significant vs. non-significant metabolites.

In the current version of the manuscript, in the Materials and Methods section and the specific paragraph on the statistical analysis of the metabolomic data, we mention the use of hierarchical clustering and significance analysis of microarrays (SAM) as the two methods used in multivariate statistical analysis. Reference 78 concerns the omic analysis software TM4/MeV about the software, and Reference 79 the SAM method, which is used for the identification of significant metabolites instead of t-test.

We agree with the Reviewer that this description may not be adequate for the reader to understand our methodology and we have now rephrased the particular section of the Methods, referring to the Methods paper of one of the authors (MK; Papadimitropoulos et al. 2018). "Untargeted GC-MS metabolomics" in *Methods in Molecular Biology* 1738:133-47 for the complete description and justification of the experimental and computational procedure and have extended the description of SAM method for the significance analysis of omic datasets, citing also a user's guide and technical document (PMID: 29654587). In addition, we have included the SAM curves for both collection time datasets in Supplementary Figures 6c and 7c.

As a metabolomics group (MK's lab), we have been using SAM and not the students' t-test in our analyses, as this is a multivariate significance analysis method developed for omic data. Omic data are not expected to follow a certain distribution (which is a pre-requisite in student's t-test and F-test) and are inter-dependent through their connectivity in the biomolecular networks. The statistical significance of the results for a particular significance threshold (called delta δ in this method) is assessed by the false discovery rate, indicating in our case the number of false positives in the group of identified as differential metabolites.

The significance threshold chosen in this study (with the corresponding FDR-median) was the strictest, for which no false positive metabolites were identified in the significant group (FDR-median equal to zero). It is noted that in SAM analysis, one does not have to set a significance threshold *a priori*, but the analysis provides the continuum of the significance thresholds and the associated FDR. This is what we state in the manuscript as “Any SAM analysis was performed using as significance threshold (δ , delta) the smallest value that corresponded to a zero false discovery rate (FDR) – median”. In the revised manuscript, we show also the additional metabolites identified as positively significant at the smallest significance threshold possible for each collection time analysis as the corresponding FDR-median is still small. These are indicated with a different color in Figures 6a,b and Supplementary Table 4, in the revised manuscript.

The authors have not measured metabolic flux. Metabolic flux is measured using heavy isotope-labeled nutrients such as ^{13}C -glucose and by performing mathematical flux analysis. Authors should avoid language such as: “Combining the differential metabolite information, we could imply an increased glycolytic flux, as supported by the increased concentration of glycerate-3-phosphate (glycerate-3P) and other downstream pathways”. The metabolomics data presented here, shows increased levels of glycolytic metabolites for tomm-40(RNAi) vs. control (although as mentioned above, metabolomics data requires more appropriate normalization), but no information about metabolic flux can be interpreted from this data.

We thank the Reviewer for this comment and we fully agree with him/her that we do not carry out fluxomics with complete balance of the examined metabolic network. However, metabolite concentrations if viewed in the context of interacting pathways from input (substrate) to output (product) can imply information about the change in the metabolic fluxes, as metabolite concentrations affect and are affected by metabolic fluxes (metabolite balancing analysis). Information about fluxes can be obtained from both metabolite and isotopomer balances as the reviewer suggests (the latter mainly for exchange fluxes in reversible reactions and fluxes in parallel pathways and metabolic cycles). It is the entire perspective that we get from the metabolic profile in combination with the implied decrease in the flux towards mitochondrial processes (due to Mito-MISS) and the observed gene expression measurements from the respective mutants (that while not showing directly the change in the flux, they indicated the potential trend in the production of the respective enzyme), that provide this potential interpretation. Knowing the limitations suggested by the reviewer, we use mild expressions (as the obtained results “could imply” a particular change in flux), as this is a legitimate interpretation supported by the available evidence.

Reviewers' Comments:

Reviewer #1:

Remarks to the Author:

The responses of the authors to the reviewer's comments from the previous version were superficial and really did not address my main concerns. One of the principle claims that mitochondrial import system suppression through tomm-40 RNAi initiates an unconventional UPRmt remains unsubstantiated by the data. First, the observation on the double RNAi of *cco-1*; *timmm-23* in a 1:9 ratio further increases lifespan compared to single RNAi of the individual gene is very likely due to a stronger activation of the UPRmt. Although the authors are aware of this issue. To exclude this possibility, they provided one piece of data showing that the expression of *hsp-6::GFP* is not further increased in *cco-1*; *timmm-23* compared to the *timmm-23* RNAi. But the expression of *hsp-6* can only represent the level of mitochondrial chaperones upon stress at most. The expression of other classical genes in the conventional UPRmt pathway should also be examined, for example, the mitochondrial chaperone (*hsp-60*) and mitochondrial proteases (e.g., *ymel-1*, *clpp-1*, *lonp-1*, etc). Second, the conclusion that MitoMISS initiates the UPRmt differently from the typical UPRmt stressors, i.e. *spg-7* RNAi, is not justified by the provided data in Fig. 4. The variation of GFP intensity within groups is quite large considering the minor differences observed between groups. Based on the previous reviews I provided, the expression of those genes should be quantified by RT-qPCR or RNA-seq, but no attempt was made to add meaningful experiments to verify these key findings. Moreover, among the nine gene expression profiles in Fig. 4j, only *icl-1* exhibits a clear differential expression upon *spg-7* RNAi versus *tomm-40* RNAi, whereas the expression of the remaining eight genes was upregulated in a similar fashion. This further confirms that the UPRmt conferred by *tomm-40* RNAi is more likely classified into the canonical UPRmt, not an unconventional pathway.

As pointed out during the initial rounds of reviews, the authors should quantify the transcripts in the main figures by RT-qPCR (Fig. 1-3, 4 and 5) instead of overly relying on the GFP expression of individual reporter strains. RT-qPCR is a relatively easy experiment that is not very time-consuming but essential to back up the conclusions they made throughout the manuscript.

The author's response to whether the proposed MitoMISS acts differently from the proteotoxic stress is unsatisfying. To make a definitive statement that uncouples the cytosolic proteotoxic stress from the MitoMISS, more evidence has to be provided, at least including the transcriptional changes of cytosolic proteostasis pathway, proteasomal activity and the level of ubiquitinated proteins. Referring to my comments in the initial rounds of reviews, this is particularly important considering what work has been done in yeast, revealing the proteotoxic stress induced by mitochondrial import deficiency. The author's speculation on why proteotoxic stress upon mitochondrial import inhibition in *C. elegans* differs from yeast are still not supported by any convincing evidence.

Most importantly, the novelty of this work is further undermined by series of papers, suggesting that the link between compromised mitochondrial protein import, UPRmt, lifespan extension and serine metabolism has already been well-known in the literature (for example: PMID: 31412237, PMID: 34252079, PMID: 27307216, PMID: 29949403).

All in all, I see no meaningful improvement in this manuscript, and I unfortunately cannot support its publication in Nature Communications.

Reviewer #3:

Remarks to the Author:

In the revised version of the manuscript, the authors have addressed most of my concerns regarding the LC-MS metabolomic analysis. The authors have repeated the metabolomics experiment and added data for the new second experiment that overall agree with the major conclusions from first metabolomic profiling. Authors have also added further explanation and supplemental figures for the statistical method used for metabolomics analysis. But there are some issues that remain to be addressed:

Minor points:

I am not convinced that authors can imply information about metabolic flux with the data presented. However, the metabolomic data in the manuscript indicates an upregulation of some of the glycolytic metabolites. Without doing any computational flux analysis, interpretation about metabolic flux should be avoided. On page 16, the authors mention "Combining the differential metabolite information, we could imply an increased glycolytic flux, as supported by the increased concentration of glycerate-3-phosphate (glycerate-3P) and other downstream pathways". It is not possible to conclude change in flux by qualitative assessment of level of some metabolites from the pathway. Metabolic flux is the passage of a metabolite through a reaction system over time, and flux analysis is the combination of time-course methodologies in metabolomics and computational modeling of pathways (see Handbook of Pharmacogenomics and Stratified Medicine, Chapter 10 – Metabolomics, Karl Burgess et al.).

Figure 6 and supplemental figures 6 and 7 have very low resolution and should be re-made with high resolution, it is very hard to read names of metabolites on the heatmaps.

It would be best if authors upload the metabolomics raw data to an online database such as MetaboLights (<https://www.ebi.ac.uk/metabolights/>)

Reviewer #4:
None

Response letter

Mitochondrial protein import determines lifespan through metabolic reprogramming and de novo serine biosynthesis

Lionaki, Gkikas, Daskalaki, Ioannidi, Klapa & Tavernarakis

Editorial and 4th Reviewer suggestions:

Following the suggestions of the Editor and the assessment of the 4th Reviewer, we toned down the claim that MitoMISS is an unconventional UPRmt paradigm. Specifically, we removed the term “unconventional UPRmt” for the MitoMISS-driven UPRmt, from the text and figure legends of our manuscript. We now simply describe the observed differences in UPRmt component activation upon MitoMISS and conventional stressors. This modification tones down the interpretation of our findings, as suggested. Nevertheless, we clearly describe the differences between MitoMISS-driven and conventional UPRmt.

In addition, we also provided a detailed point-by-point response to the comments of the other Reviewers, below. Moreover, beyond what was asked by the Reviewers, we now included two new control experiments, which highlight the effects of inhibiting de novo serine biosynthesis on mitochondria (the interplay between MitoMISS and de novo serine biosynthesis). As shown in the new Supplementary Fig 10a-c, phgdh-1 suppression had minimal effects on mitochondria. More importantly, TOMM-40 depletion upon phgdh-1 RNAi recapitulates the MitoMISS phenotype with respect to mitochondrial abundance and functionality, despite the fact that it loses its beneficial effects on lifespan.

Reviewer 1:

The responses of the authors to the reviewer’s comments from the previous version were superficial and really did not address my main concerns. One of the principle claims that mitochondrial import system suppression through tomm-40 RNAi initiates an unconventional UPRmt remains unsubstantiated by the data. First, the observation on the double RNAi of cco-1;timm-23 in a 1:9 ratio further increases lifespan compared to single RNAi of the individual gene is very likely due to a stronger activation of the UPRmt. Although the authors are aware of this issue.

We respectfully disagree with the Reviewer. We certainly did not provide superficial responses to the comments we received. On the contrary, we have done all specific experiments requested, and much more. In addition, as can clearly be seen in the previous response letters, we always provided detailed, point-by-point responses, including numerous additional experimental data, to all specific comments of the

Reviewer. Our manuscript has now grown to contain 17 complex figures (7 main and 10 supplementary figures), comprising 124 panels in total, as well as, 4 extended supplementary tables. With regard to this new comment of the Reviewer, it is known that strong UPR^{mt} activation may also have negative results on lifespan, as is the case in constitutively active *atfs-1* mutants. Moreover, it is known that strong depletion of ETC components has negative effects on lifespan (PMID 17914900). Specifically, for *cco-1* RNAi we used exactly the dilution that has been previously published to extend lifespan (1/10) and combined it with *timm-23* RNAi to look for epistatic effects on lifespan and UPR^{mt} activation. As we discuss in the manuscript, combination of *cco-1* and MitoMISS produce an additive effect on lifespan, but not on *hsp-6* activation, corroborating the notion that it is not the level of mitochondrial chaperone activation that is critical for MitoMISS longevity.

With regard to the reviewer's objection on the non-canonical axis of UPR^{mt}, and given that the findings about the differential features of MitoMISS-driven UPR^{mt} are not the main focus of our manuscript, we decided to remove the term "non-conventional" from the main text and figure legends and only describe the data highlighting the observed differences. Moreover, we moved Figure 4 to the Supplementary materials (now new Supplementary Figure 5).

To exclude this possibility, they provided one piece of data showing that the expression of *hsp-6::GFP* is not further increased in *cco-1;timm-23* compared to the *timm-23* RNAi. But the expression of *hsp-6* can only represent the level of mitochondrial chaperones upon stress at most. The expression of other classical genes in the conventional UPR^{mt} pathway should also be examined, for example, the mitochondrial chaperone (*hsp-60*) and mitochondrial proteases (e.g., *ymel-1*, *clpp-1*, *lonp-1*, etc).

We have followed the same approach that is widely used in the literature to assess UPR^{mt} in *C. elegans*. See for example, a very recent paper published in *Nature Communications*:

Shpilka T, et al. UPR^{mt} scales mitochondrial network expansion with protein synthesis via mitochondrial import in *Caenorhabditis elegans*. *Nat Commun.* 2021; 12: 479.

Numerous others are listed here: PMID: 32510480, and here: PMID: 29424373

We would like to note here that the Reviewer did not propose a specific experiment towards this direction, in the previous revision rounds. By contrast, we have experimentally addressed all previous concerns raised, in all cases.

Second, the conclusion that MitoMISS initiates the UPR^{mt} differently from the typical UPR^{mt} stressors, i.e. *spg-7* RNAi, is not justified by the provided data in Fig. 4. The variation of GFP intensity within groups

is quite large considering the minor differences observed between groups. Based on the previous reviews I provided, the expression of those genes should be quantified by RT-qPCR or RNA-seq, but no attempt was made to add meaningful experiments to verify these key findings. Moreover, among the nine gene expression profiles in Fig. 4j, only *icl-1* exhibits a clear differential expression upon *spg-7* RNAi versus *tomm-40* RNAi, whereas the expression of the remaining eight genes was upregulated in a similar fashion. This further confirms that the UPR^{mt} conferred by *tomm-40* RNAi is more likely classified into the canonical UPR^{mt}, not an unconventional pathway.

The RNA-seq analysis was proposed by us in our last response letter, for a follow up study, and was not asked by the Reviewer. Instead, the Reviewer had asked for RT-qPCR experiments. We have provided RT-qPCR results for nine additional genes to corroborate our initial findings. We need to note at this point that RT-qPCR and RNA-seq does not work for worms as it works for cells or tissues. In worms, RNA is unavoidably extracted from whole animals, containing heterogeneous tissues and many different cell types, all of which are physiologically diverse and, thus, not expected to respond in the same way. Consequently, the overall response can be masked in a biochemical assay. Transcriptional reporters have the advantage over RT-qPCR, that they can highlight transcriptional responses in the tissue of interest. Regarding the non-conventional UPR^{mt}; UPR^{mt} can be induced by a variety of stimuli. This is very clearly demonstrated by the work of Auwerx and colleagues (PMID 28566324). Not all of them are expected to elicit the exact same transcriptional response. Similarly, previous studies on UPR^{ER} activation upon various ER stress inducers have shown that different subset of transcripts are expressed based of the nature of the stimuli (PMID 30333136). Therefore, an RNA-seq profile of *spg-7* and *tomm-40* RNAi treated animals is not appropriate and is not expected to provide meaningful information. It might even be misleading.

This is not surprising. In our previous submission, the main reason we stated that MitoMISS induces a non-conventional UPR^{mt} is the fact that three key UPR^{mt} players are either not induced or not required for MitoMISS-related lifespan extension (*DVE-1*, *HAF-1*, *HSP-60*) (PMIDs 32934238, 20188671) (Figure 3f,g and Supplementary Figure 4g-j). Nevertheless, we decided to remove the characterization “non-conventional” from the main text and figure legends and just describe the data highlighting the observed differences. In addition, we moved Figure 4 to the Supplementary Material (currently, Supplementary Figure 5). As we have noted, more work is required to decipher the link between *spg-7* and MitoMISS driven UPR^{mt}, and this is definitely within our future plans. However, this is not within the scope of this paper.

As pointed out during the initial rounds of reviews, the authors should quantify the transcripts in the main figures by RT-qPCR (Fig. 1-3, 4 and 5) instead of overly relying on the GFP expression of individual reporter strains. RT-qPCR is a relatively easy experiment that is not very time-consuming but essential to back up the conclusions they made throughout the manuscript.

We have already provided RT-PCR verifications for all key findings of our study as it has been also

proposed by the 2nd Reviewer, during the first round of revision (Supplementary Fig.1c, Supplementary Fig.3d, Supplementary Fig.4c, d, new Supplementary Fig. 5j, new Supplementary Fig. 6a,b,d and Supplementary Table 1). The 2nd Reviewer was fully satisfied by our responses and suggested publication without further comments.

The author's response to whether the proposed MitoMISS acts differently from the proteotoxic stress is unsatisfying. To make a definitive statement that uncouples the cytosolic proteotoxic stress from the MitoMISS, more evidence has to be provided, at least including the transcriptional changes of cytosolic proteostasis pathway, proteasomal activity and the level of ubiquitinated proteins. Referring to my comments in the initial rounds of reviews, this is particularly important considering what work has been done in yeast, revealing the proteotoxic stress induced by mitochondrial import deficiency. The author's speculation on why proteotoxic stress upon mitochondrial import inhibition in *C. elegans* differs from yeast are still not supported by any convincing evidence.

As we explicitly explained in our previous response letter, the revised version of our manuscript contains RT-qPCR data showing that MitoMISS does not cause induction of transcriptional changes of cytosolic or ER proteotoxic pathways (Supplementary. Fig. 3d) (please, see our last response letter). Indeed, these findings are in agreement with other reports of mitochondrial protein import inhibition in nematodes (PMID: 29117555, 27610574). While our manuscript was under review, another paper has been published in PLoS Biol (PMID 34252079), regarding the role of mitochondrial protein import in cytosolic proteotoxicity (also referred to by the Reviewer, in their last comment, below). In this paper, *dnj-21* has been genetically suppressed to inhibit mitochondrial import. DNJ-21 is a component of the PAM motor that cooperates with TIM23 complex to import proteins into the matrix. It is evident that DNJ-21 depletion does not block mitochondrial import as its effects on mitochondrial function and morphology are minimal. However, they show that *dnj-21* depletion in *C. elegans* leads to lifespan extension, although in an ATFS-1-independent manner. Moreover, they show that *dnj-21* depletion activates some features of the UPR^{mt} highlighting selected targets (*hsp-6* is induced but not *dnj-10* or *timm-23*). This is in agreement with our findings that MitoMISS activates some but not all key UPR^{mt} players.

Interestingly, *dnj-21* suppression although slightly activated proteasomal activity, as shown by the newly established UbG76V-Dendra2 strain, however it didn't activate transcriptional induction of proteasomal subunits, protein levels of proteasomal subunits and it didn't alter assembly of proteasomal subunits. It required a very specific assay to identify a rather mild difference in proteasomal activity. Moreover, UPR^{ER} components were not transcriptional activated, again in agreement with our MitoMISS related findings and in contrast to the published yeast results. The data presented in this paper are essentially in agreement with our findings, that a robust proteostatic response in the cytoplasm or the ER, similar to the one described in budding yeast, is not induced in *C. elegans*. This actually alleviates the

concerns of the Reviewer.

Most importantly, the novelty of this work is further undermined by series of papers, suggesting that the link between compromised mitochondrial protein import, UPR^{mt}, lifespan extension and serine metabolism has already been well-known in the literature (for example: PMID: 31412237, PMID: 34252079, PMID: 27307216, PMID: 29949403). All in all, I see no meaningful improvement in this manuscript, and I unfortunately cannot support its publication in Nature Communications.

Respectfully, we do not agree with the Reviewer. We are well aware of these papers, and our study is distinctly different (please, also see our response to the previous comment, above). Indeed, our manuscript reports the following novel, key findings:

1. We establish MitoMISS as a novel longevity paradigm independent of caloric restriction, low insulin signaling, mild mitochondrial dysfunction or mitohormesis. Interestingly, pharmacological targeting of mitochondrial protein import and associated reduction of mitochondrial abundance also leads to longevity which generalizes the effects of MitoMISS on lifespan.
2. MitoMISS activates features of UPR^{mt}. We have now incorporated new data showing that mRNAs of ATFS-1 targets genes are differentially regulated upon *spg-7* depletion (typical UPR^{mt} stressor) and MitoMISS. Our data suggest that different mitochondrial stressors elicit distinct UPR^{mt} responses, acting in parallel to control longevity.
3. MitoMISS does not cause robust proteotoxic stress outside mitochondria. Specifically, new qRT-PCR data confirm the lack of a robust proteotoxic stress response in the endoplasmic reticulum or in the cytosol.
4. ATFS-1-dependent induction of mitochondrial chaperones does not always correlate nor is required for longevity. In contrast, MitoMISS rescues the toxicity of mitochondrial chaperone depletion. This intriguing new finding suggests that reduction of mitochondrial load rescues animals from severe mitochondrial stress.
5. MitoMISS induces glucose uptake and glycolysis in an ATFS-1-dependent manner. Genetic analysis and metabolomic profiling revealed that MitoMISS induces accumulation of glycolytic intermediates that are now channeled to anabolic pathways branching from glycolysis like *de novo* serine biosynthesis and glycerophospholipid biosynthesis. Notably, inhibition of either glycolysis or *de novo* serine biosynthesis abolishes the beneficial effects of MitoMISS on lifespan, satisfying a causative role of metabolic reprogramming to longevity.
6. Redirecting carbon sources to alternative metabolic routes upon MitoMISS alleviates glucotoxicity in the context of organismal lifespan.

7. Tissues that convey the benefits from MitoMISS on lifespan are the body wall muscles and the intestine, while neuron- and hypodermis-specific MitoMISS fail to promote longevity.

Overall, the main conclusion of our extensive study is that the mitochondrial import system, through UPR^{mt} activation, adjusts cellular metabolism towards *de novo* serine biosynthesis *in vivo*. Thus, our manuscript reports original findings that provide significant insight, relevant to the metabolic alterations that promote or undermine survival and longevity, in response to mitochondrial perturbations. Our study reveals a novel molecular mechanism by which modulation of cellular mitochondrial content triggers metabolic adaptations that lead to lifespan extension and amelioration of glucotoxicity in the context of a whole organism. *De novo* serine biosynthesis lies in the center of this metabolic switch and characterizes this novel longevity paradigm. To our knowledge, this is the first time that the metabolome of UPR^{mt} has been analyzed in invertebrates. This metabolic adaptation leads to lifespan extension and amelioration of glucotoxicity, in the context of a whole organism. Moreover, our study suggests that the metabolic reprogramming upon mitochondrial stress is evolutionarily conserved. Finally, our data extend the pro-longevity effects of UPR^{mt}, from the currently established idea of sustaining mitochondrial proteostasis to the coordination of the respective metabolic adaptations, a notion that, until today, has not been addressed in any publication.

Minor points:

Reviewer 3:

I am not convinced that authors can imply information about metabolic flux with the data presented. However, the metabolomic data in the manuscript indicates an upregulation of some of the glycolytic metabolites. Without doing any computational flux analysis, interpretation about metabolic flux should be avoided. On page 16, the authors mention “Combining the differential metabolite information, we could imply an increased glycolytic flux, as supported by the increased concentration of glycerate-3-phosphate (glycerate-3P) and other downstream pathways”. It is not possible to conclude change in flux by qualitative assessment of level of some metabolites from the pathway. Metabolic flux is the passage of a metabolite through a reaction system over time, and flux analysis is the combination of time-course methodologies in metabolomics and computational modeling of pathways (see Handbook of Pharmacogenomics and Stratified Medicine, Chapter 10 – Metabolomics, Karl Burgess et al.).

We thank the reviewer for acknowledging our efforts to address his concerns and we are happy meet his expectations. We appreciate the reviewer’s comment on avoiding interpretation on metabolic flux and we have now changed this statement in our revised manuscript according to his/her suggestion.

Figure 6 and supplemental figures 6 and 7 have very low resolution and should be re-made with high resolution, it is very hard to read names of metabolites on the heatmaps.

We thank the reviewer for the comment. This issue arises from the conversion of the original PPT files to TIFF files, by the online manuscript submission system. We have submitted the high-resolution PPT files of Figures 6, S6 and S7 (now new Figure 5 and Supplementary Figure 7 and 8), as suggested.

It would be best if authors upload the metabolomics raw data to an online database such as MetaboLights (<https://www.ebi.ac.uk/metabolights/>).

Of course, we fully agree with the Reviewer's suggestion. We do intend to do so, as soon as our manuscript has been accepted for publication.

Reviewers' Comments:

Reviewer #4:

Remarks to the Author:

The authors have addressed my concerns.